# A sticky Poisson Hidden Markov Model for solving the problem of over-segmentation and rapid state switching in cortical datasets

Tianshu Li[1,2,3], Giancarlo La Camera [1,2,3]*

**1** Department of Neurobiology & Behavior, Stony Brook University, Stony Brook, NY, United States, **2** Graduate Program in Neuroscience, Stony Brook University, Stony Brook, NY, United States, **3** Center for Neural Circuit Dynamics, Stony Brook University, Stony Brook, NY, United States

* giancarlolacamera@stonybrook.edu

## Abstract

The application of hidden Markov models (HMMs) to neural data has uncovered hidden states and signatures of neural dynamics that are relevant for sensory and cognitive processes. However, training an HMM on cortical data requires a careful handling of model selection, since models with more numerous hidden states generally have a higher likelihood on new (unseen) data. A potentially related problem is the occurrence of very rapid state switching after decoding the data with an HMM. The first problem can lead to overfitting and over-segmentation of the data. The second problem is due to intermediate-to-low self-transition probabilities and is at odds with many reports that hidden states in cortex tend to last from hundred of milliseconds to seconds. Here, we show that we can alleviate both problems by regularizing a Poisson-HMM during training so as to enforce large self-transition probabilities. We call this algorithm the 'sticky Poisson-HMM' (sPHMM). The sPHMM successfully eliminates rapid state switching, outperforming an alternative strategy based on an HMM with a large prior on the self-transition probabilities. When used together with the Bayesian Information Criterion for model selection, the sPHMM also captures the ground truth in surrogate datasets built to resemble the statistical properties of the experimental data.

## Introduction

The Hidden Markov model (HMM) [1] is a dynamical system undergoing randomly timed transitions among unobserved (hidden) states that generate specific observations (in this paper, the observations are Poisson spike trains emitted by an ensemble of spiking neurons). In data analysis, the HMM can be used as an unsupervised method for segmenting a time series into discrete states. With this goal, HMMs have been widely applied to phoneme segmentation in speech recognition, DNA sequence analysis, behavioral analysis, and many other problems [2–4]. In neuroscience, the HMM has been used successfully to uncover sequences of hidden states characterized by vectors of firing rates across simultaneously

**Data availability statement:** Custom computer code to implement the HMM algorithms studied

in this article is freely available at https://github.com/lacameralab/HMM-spikes

**Funding:** NIH/NINDS Brain Initiative (1UF1NS115779) to G.L.C.

**Competing interests:** The authors have declared that no competing interests exist.

recorded neurons, although BOLD signals [5] and other signals related to neural activity have also been analyzed. These studies have shown that cortical neural activity often unfolds through sequences of discrete, metastable states [6–17], a phenomenology that can be reproduced in model networks of spiking neurons with a clustered architecture [13,18–20]; for reviews, see [21,22]. After early seminal work in the nineties [6–8], the HMM and its variations are now frequently used to uncover sequences of metastable states in ensembles of spikes trains and other types of neural activity.

We show an example of such an analysis in Fig 1A. The top panel shows the raster plots of 5 spike trains simultaneously recorded from 5 neurons of the gustatory cortex of an alert rat [23]. Each line is a spike train and each vertical segment marks the time at which a spike was recorded (spike rasters). This spatiotemporal pattern of activity was segmented with an HMM; the model was first trained on the full dataset (comprising many trials) to infer the

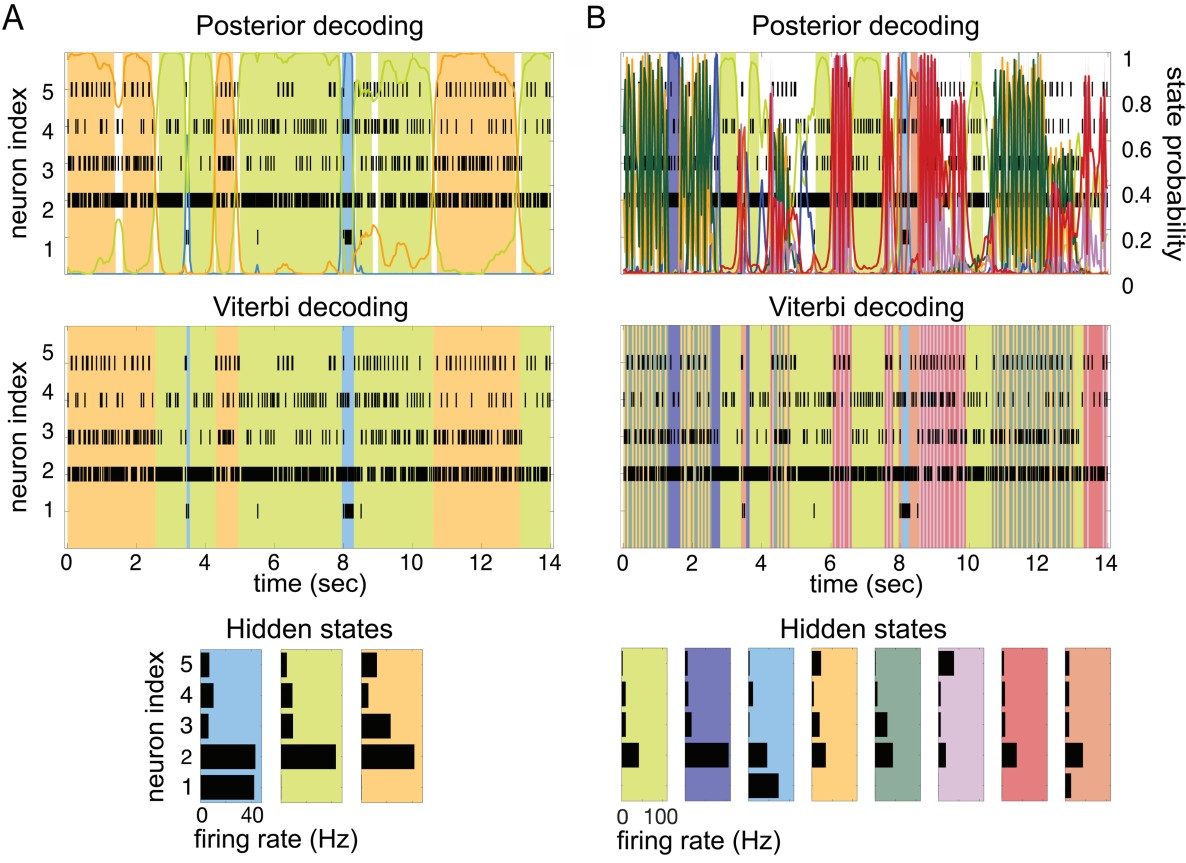

**Fig 1. HMM of spike data and the problem of rapid state switching. A:** An example of HMM decoding of a sequence of spike trains recorded simultaneously from 5 neurons of the primary gustatory cortex of alert rats [23]. The rasters show the spike times (vertical segments) for each spike train. The color-shaded areas are the hidden states decoded from an HMM previously fit to different trials of the same dataset. An example of posterior decoding (top) and Viterbi decoding (middle) are shown (see the text for details). The HMM used to fit and decode the data had 3 hidden states (bottom panel), defined as firing rate vectors across the simultaneously recorded neurons. The colored curves in the top panel represent the posterior probability of being in the corresponding hidden state given the data. Shaded segments are portions of the data where the largest posterior probability among the states was larger than 0.8. As in many cortical datasets [21,22], the hidden states have durations varying between a few hundred milliseconds to a few seconds. **B:** Decoding the same data of panel A with 8 hidden states instead of 3 leads to highly frequent state transitions (rapid state switching). The algorithms used were the sPHMM with $m = 3$ hidden states (in A) and the PHMM with $m = 8$ (in B). Fitting and decoding an HMM require discretizing time in bins; time bins of 50 ms were used in both panels. See Sec. "Algorithms" for details.

model parameters, and then used to decode the neural activity in unseen trials, i.e., to segment the neural activity into discrete and distinct patterns called hidden states. The decoded segments are marked by color-shaded areas denoting the presence of hidden states (we give later full details on how this analysis is conducted). The hidden states are vectors of stationary firing rates across the neurons, and are called hidden because they are not directly observable; in this example, 3 hidden states were found and are shown in the bottom panel Fig 1A (in color matching the shadings in the top 2 panels). Two main procedures are routinely used to decode the neural activity with an HMM: posterior decoding and Viterbi decoding, shown in the top two panels. In posterior decoding (top panel), one assigns each bin of data to the hidden state with the largest posterior probability; in Viterbi decoding (middle panel), one assigns the states so as to maximize the probability of the entire sequence of states. These two methods often give the same results, as shown in Fig 1A. The white spaces in the top panel reflect our choice of not assigning bins to states if no state has a posterior probability larger than 0.8 [10].

To successfully fit an HMM to spike data one must correctly select the number of hidden states. In the frequentist approach typically used when fitting an HMM, a number of hidden states is decided a priori, and the best model emerges from the comparison of models with different numbers of states. Too large a number of states can lead to overfitting, 'over-segmentation' and rapid state switching, i.e., the creation of redundant states and the frequent switching among them (much more frequent than it can be reasonably expected from the nature of the data). Too small a number of states can instead lead to a poor fit, where genuinely different states are collated into the same state, and transitions among some of the true states are missed.

Selecting the correct number of hidden states is complicated by the fact that the log-likelihood of the model given new data does not decrease when increasing the number of hidden states, as we will show here in a systematic manner. In such a situation, one cannot simply select the model with the largest likelihood, because this typically would result in over-segmentation. A related problem is rapid state switching (Fig 1B). This phenomenon occurs when oscillations in state decoding occur unreasonably fast given the data; as illustrated in Fig 1B, this may occur regardless of the decoding method used. Reasonable state durations in this system are 300–1000 ms [10,13], yet the model finds frequent transitions among states of extremely short duration. Note that rapid state switching of this sort is surely incorrect for spike data: in cortex, spike trains are erratic and resemble a Poisson process, which means that large variations in the interspike intervals are to be expected even in periods of constant firing rate. It is apparent from Fig 1B that many state transitions occur over times so short that no significant changes can realistically occur (or be detected) in the underlying spike trains.

We will show that rapid state switching occurs as a consequence of intermediate-to-low self-transition probabilities in the underlying Markov chain, and is more likely to occur in the presence of many states. In fact, we show in this paper that given a large number of states, the training algorithm will seldom converge to a solution with large self-transition probabilities. On the other hand, training with a small number of states does not guarantee the absence of rapid state switching. Hence, even when using a suitable (not too large) number of states, care must be taken to avoid self-transition probabilities that are too small to be compatible with the data. Although some of these problems may be thought of as the consequence of slowly changing firing rates in some neurons, we emphasize that fast transitions among discrete states compatible with an HMM description (rather than, for example, a collection of inhomogeneous Poisson processes) have been demonstrated in several cortical areas, including the prefrontal [8] and the motor and premotor [12] cortex of monkeys and the gustatory

cortex of rats [10]. As illustrated in Fig 1, in such datasets the problem of rapid state switching (panel B) is a consequence of a bad training algorithm rather than incompatibility with an HMM (panel A).

In this paper we present a way to train an HMM so as to keep the self-transition probabilities above a desired threshold, an algorithm that we call 'sticky Poisson-HMM' (sPHMM). We compare this method with a vanilla Poisson-HMM (PHMM) and a Poisson-HMM with a Dirichlet prior (DPHMM) on the self-transition probabilities. The DPHMM encourages high self-transition probabilities during training but, unlike the sPHMM, does not impose a hard threshold on them. We test and compare these algorithms on cortical datasets as well as surrogate datasets for which ground truth is known. Our main conclusion is that the use of the sPHMM together with the well known Bayesian Information Criterion to select the number of hidden states, is very successful at preventing both overfitting and rapid state switching, and offers a good practical solution to model selection.

To facilitate a straightforward application of our methods, we provide a detailed description of the algorithms together with custom computer code.

## Results

### HMM analysis of spike data

Before presenting our main results, we summarize in this section the procedure of fitting an HMM to spike data and the main issues that motivated this study. An HMM is characterized by $m$ hidden states $S_1, ..., S_m$, a matrix $\Gamma$ of transition probabilities $\gamma_{ij}$ from state $i$ to state $j$, and a probability distribution $e_i(O)$ over the observations that can be made in each state: when in state $S_i$, the model emits an observation $O$ according to $e_i(O)$. The latter is called the 'emission probability' or the 'state-dependent distribution'. Technically, the initial distribution of the states is also required to fully characterize the model; however, in this paper the initial distribution has no important role, and we only deal with it when discussing the details of the algorithms.

An HMM remains in state $S_i$ for an (exponentially distributed) random time $t_i$, where it emits an observation $O$ with probability $e_i(O)$, after which it transitions to a new state, say state $S_j$, with probability $\gamma_{ij}$. The HMM remains in the same state with probability $\gamma_{ii}$ (the 'self-transition' probability). Both the transition probability $\gamma_{ij}$ and the emission probability $e_i(O)$ depend only on the current state $S_i$ and not on previous history (the Markov assumption). In our case, we assume that each state $S_i$ is a vector of firing rates $\{\lambda_{ni}\}_{n=1,...,N}$ across $N$ simultaneously recorded neurons, while each observation $O$ while in state $S_i$ is a vector of spike counts across the same $N$ neurons, interpreted as a noisy observation of the $\{\lambda_{ni}\}$. The goal of fitting an HMM to data is to infer the number of hidden states $m$, the emission probabilities $e_i(O)$, and the transition probability matrix $\Gamma$. In Poisson-HMMs (which are the focus of this work), $e_i(O)$ depends parametrically on the firing rates $\lambda_{ni}$ defining the hidden states, hence inferring the state-dependent distributions $e_i$ also amounts to inferring the hidden states themselves (see Materials and Methods).

As in most works mentioned in the Introduction, we use a maximum likelihood approach to model selection. This means that we select the HMM that best describes the data by comparing the log-likelihood (LL) of models with different numbers of states. First, an HMM is fitted to the data assuming a given number of hidden states $m$, where $m = 2, ..., M$. For each $m$, the log-likelihood of the model given the data, $LL(m)$, is computed and then the best model is chosen. This is ideally done by comparing models on a validation set not used for training. On the validation set, the LL is expected to increase at first (as the number of states increases), and then decrease due to overfitting by models with large $m$ (see e.g. [24]). In between, one

expects a maximum of the LL in correspondence of an intermediate value $m^*$, which gives the best model (i.e., the model with the optimal number of hidden states). Once the best model is selected, it can be used to make further inference on the data, such as obtaining a sequence of 'decoded' states in each trial (see e.g. Fig 1A), or associating a hidden state with a specific task variable (see e.g. [15,17,25], for a few examples).

As mentioned in the Introduction, two problems often emerge when using the strategy outlined above: (i) the log-likelihood of the model on the validation set does not attain a clear maximum in correspondence of the optimal number of states, which makes the model selection problematic; (ii) the selected model produces extremely frequent transitions during decoding which appear as very rapid state switching (see Fig 1B); this is at odds with expected state durations lasting from hundreds of milliseconds to seconds [21], as emphasized in the Introduction.

The first problem (model selection) is a typical problem of HMM analysis; we show examples later in Sec. "Model selection". In brief, imposing a larger number of hidden states than required does not decrease the LL of the model on validation sets. This can be understood from the fact that the HMM, to accommodate additional states, can always infer noisy copies of a given state during training, and allow transitions among those copies. The second problem is related to self-transition probabilities that are not large enough. Rapid state switching and, in general, the inadequate modeling of the temporal persistence of hidden states, is also known in other domains and can occur in more sophisticated HMMs such as the Hierarchical Dirichlet Process HMM [26]. Intermediate-to-low values of the self-transition probabilities are the culprit, because these probabilities control the typical amount of time spent in the same state (see S1 Appendix). For example, the authors of [27] have reported rapid state switching when initializing their diagonal elements to a value 0.5 or to random initial values (see their Fig 3E-F); however, the problem may also occur when starting from high initial values, as it occurs in our examples. Although not necessary to generate rapid state switching, over-segmentation creates an opportunity for it by increasing the number of possible state transitions: since the sum of transition probabilities from state $S_i$ to any other state $S_j$ must add up to 1, some self-transition probabilities will converge to values small enough to make rapid state switching more likely. It is intuitive that an algorithm which encourages large values of the self-transition probabilities during training should help to avoid this problem. We present two such algorithms below.

## Algorithms

We tested three algorithms for training Poisson HMMs on both experimental and simulated datasets. The datasets were: (i) experimental data from simultaneously recorded cortical spike trains (EXP); (ii) surrogate data generated by a spiking neural network (SNN); and (iii) surrogate data generated by a Markov-modulated Poisson process (MMPP), for which an HMM is the true model and ground truth is known (full details in Materials and Methods). The experimental data comprised ensembles of simultaneously recorded spike trains from the rat brain (medial prefrontal cortex, orbitofrontal cortex, gustatory cortex, gustatory thalamus and basolateral amygdala; see Materials and Methods). The results were similar across experimental datasets; for concreteness, in this paper we illustrate our results for the data obtained in the gustatory cortex. The SNN data was generated by a spiking network with a clustered architecture producing metastable dynamics similar to that observed in the experiments [13].

The HMMs trained on these datasets were three variations on training a Poisson-HMM: i) a 'multivariate' Poisson-HMM (PHMM), ii) a Poisson-HMM trained with a Dirichlet prior over the transition probabilities (DPHMM), and iii) a Poisson-HMM trained with

a hard lower bound on the self-transition probabilities, which we call the 'sticky Poisson-HMM' (sPHMM; see Materials and Methods for full details on the algorithms). The DPHMM encourages large values of the self-transition probabilities during training, while the sPHMM enforces the self-transition probabilities to be greater than a threshold, here set equal to 0.8 (this value here corresponds to mean state durations $\geq$ 250 ms, see S1 Appendix for details). Examples of segmentation obtained with the DPHMM and sPHMM are shown in Fig 2.

For comparison, in the right-most column of Fig 2 we also show the performance of an HMM with categorical observations. This model has a finite number of observations for each state, each observation corresponding to only one neuron firing a single spike in the current spike bin, and therefore we call it the Multinoulli-HMM (MHMM); see Materials and Methods for details. The MHMM has been used often for fitting spike data [10,12,13,15,17] and therefore is a good benchmark for our Poisson-HMMs. In fact, the MHMM can be interpreted as an alternative way to fit a Poisson-HMM to spike data, because the MHMM approximates well the Poisson-HMM for very short bins, low firing rates, and low numbers of neurons (see Materials and Methods). Although the MHMM offers reliable performance, this method is not immune to the problems of over-segmentation and rapid state switching. Moreover, the MHMM is computationally expensive due to the small time bin required (typically, not greater than 5 ms), and for these reasons in the following we focus on the analysis of Poisson-HMMs.

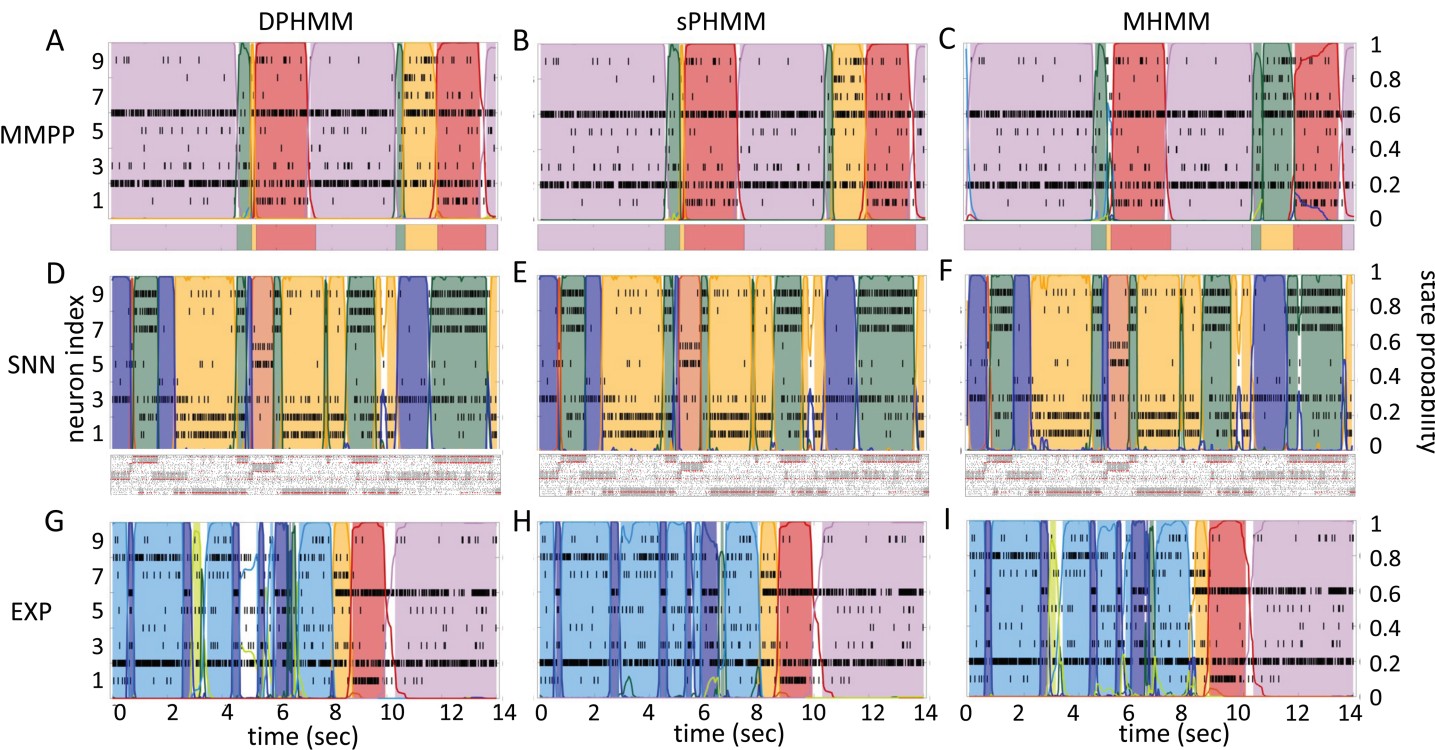

**Fig 2. Examples of three HMMs on three spike datasets.** From left to right, examples of PHMM with Dirichlet prior (DPHMM), sticky-PHMM (sPHMM) and Multinoulli-HMM (MHMM) on three datasets (MMPP, SNN, EXP; see the text for details) with spike trains from 9 simultaneously recorded neurons. **A-C)** MMPP dataset. Top panels: single trial decoding with $m = 8$ hidden states for all three HMMs. This dataset was generated starting from the EXP dataset in panels G-I (see Materials and Methods for details). Bottom panels: the true state and state transitions used to generate the spike data. **D-F)** SNN dataset. Top panels: single trial decoding with $m = 4$ hidden states for all algorithms. Bottom panels: raster plot of 5 clusters of the original spiking neural network from which the 9 neurons in the top panels were chosen from (red dots; see Materials and Methods for details). **G-I)** EXP dataset. Single trial decoding with $m = 8$ hidden states in all cases.

## Model selection

The HMMs were fit to the data assuming each time a given number of hidden states, $m$, and the optimal $m$ was chosen by maximum likelihood. We compared three criteria for selecting $m^*$, the optimal value of $m$. One criterion, cross-validation, selects the model based on performance (LL) on validation sets not used for training; the other two criteria use the LL on the training set, $LL_t$, but penalize the LL based on model complexity and the criterion used; specifically, the selected model is the one minimizing either the Bayesian Information Criterion (BIC) or the Akaike Information Criterion (AIC), defined as

$$\mathrm{BIC}(m) = -2LL_t(m) + K \ln D, \tag{1}$$

$$\mathrm{AIC}(m) = -2LL_t(m) + 2K, \tag{2}$$

where $K$ is the number of parameters tuned during training and $D$ is the total number of observations in each session (i.e., the total number of time bins across all trials). Note that, in general, $m^*$ takes different values for different criteria.

Fig 3 shows the LL curves on the validation sets, $LL_v(m)$, and the BIC and AIC scores as a function of $m$ for the PHMM trained on three different datasets. As anticipated, the LL monotonically increases, or plateaus, without a clear maximum. Selecting the largest $m$ would certainly lead to overfitting, and a different strategy is needed. Possible strategies are suggested by the visible 'elbow' in each $LL_v$ curve (red ellipses). Roughly, the points in the elbow region strike a good compromise between model complexity and generalization performance, because they correspond to parsimonious models for which $LL_v(m)$ is nevertheless close to its maximum. But how can one extract the best value of $m$ according to this strategy? In Fig 3A, where a near-constant plateau is clearly delineated, one could select the largest number of states at the beginning of the plateau (e.g., $m^* = 10$ or $m^* = 11$ in Fig 3A). In this case this choice is rather accurate (the true $m^* = 10$). The situation is trickier in datasets produced by a metastable spiking network (Fig 3B) or collected experimentally (Fig 3C). In those cases, the elbow region is less clearly defined and the $LL_v$ curve is monotonically increasing for all tested values of $m$. Different algorithms designed to single out the optimal number of states in this case tend to produce different answers; we defer the analysis to Sec. "Model selection on ground truth datasets".

The AIC curves tend to mirror the $LL_v$ curves; in some cases, a shallow but clear minimum is obtained (Fig 3B-C). In contrast, the BIC curves tend to have a clear minimum that is compatible with the elbow region of the corresponding $LL_v$ curve. Except for very small datasets or large numbers of parameters (which is not the case here), AIC penalizes less the LL and therefore provides a larger $m^*$ than BIC.

Similar results were observed for the DPHMM and the sPHMM algorithms, as shown in Figs 4-5. However, having a prior in the DPHMM offers a way to overcome the problem of a monotonic $LL_v$ curve, because in this case we have the option of cross-validating the log-posterior $LP_v(m)$ rather than the LL. The log-posterior equals the sum of the LL and the log-prior on the transition probabilities; by comparing the log-posteriors on the validation set, we express a preference for models that are compatible with the prior, i.e., having higher self-transition probabilities. Since the log-prior is negative and depends on $m$ (Materials and Methods, Sec. "DPHMM"), this results in a non-monotonic behavior of the $LP_v$ curves, with a clear maximum that coincides with, or is very close to, the minimum of the corresponding BIC curve (see the green curves in the leftmost plots of Fig 4). Improved results are also obtained when using AIC, whereas BIC results are unaltered (the BIC and AIC scores were

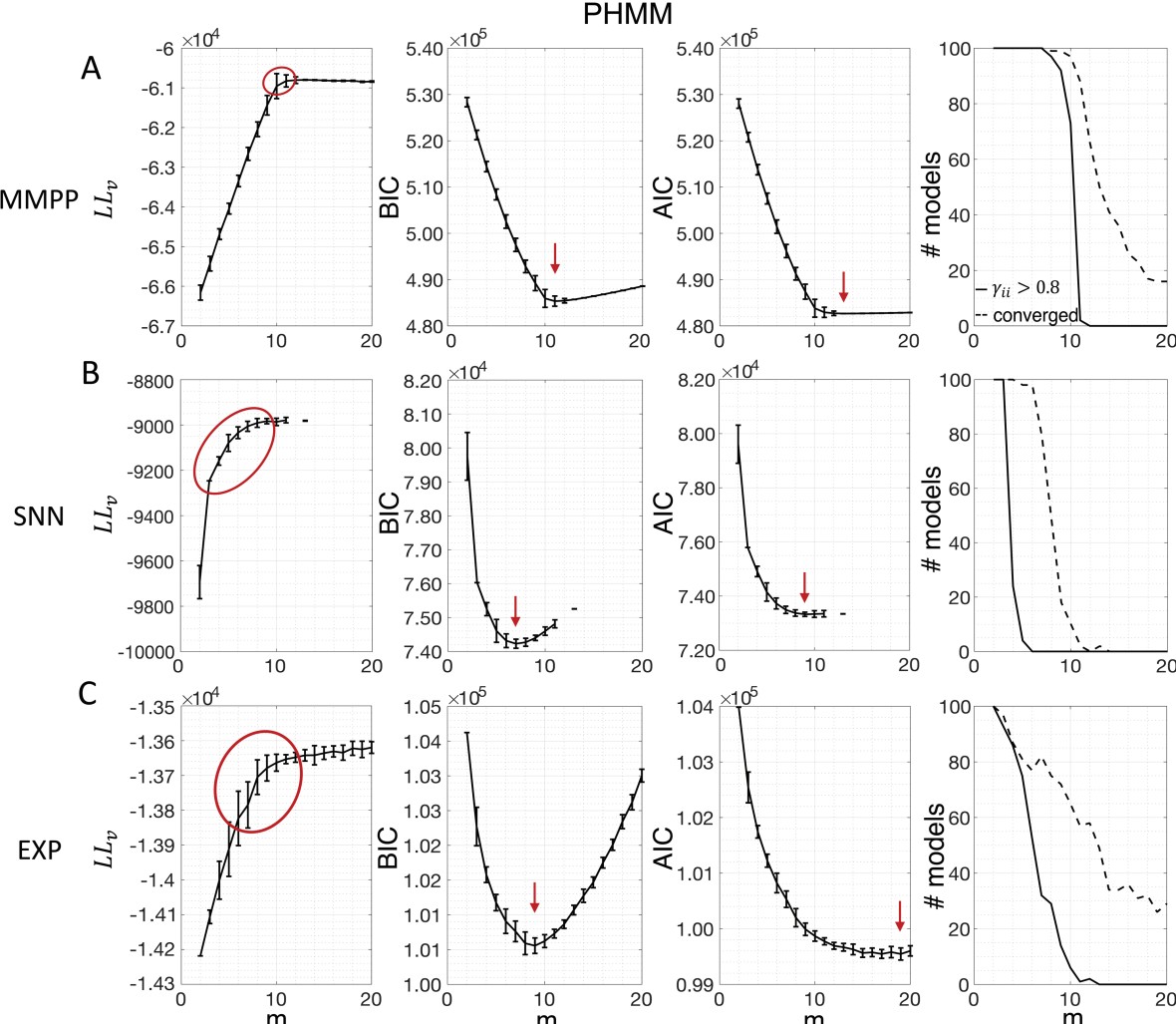

**Fig 3. Examples of model selection methods for the PHMM. A:** Results for an MMPP dataset comprising 20 neurons, 50 trials, 14 s per trial, and 10 states. *Left panel:* LL on validation sets as a function of *m*. The black curves show the average log-likelihoods (across initial guesses) of all models whose algorithm converged to a solution during training (vertical bars display one standard deviation across 100 different initial guesses for the parameter values). The region encircled by a red curve indicates the "elbow" region presumably containing the optimal model. Any *m* within this range can be a candidate for the optimal number of states. *Middle-left panel*: BIC scores averaged across initial guesses as a function of *m*. Black curves and vertical bars show mean and standard deviation for each *m*. Red arrow: minimum of average BIC. *Middle-right panel*: same as middle-left panel, but showing the AIC scores. *Rightmost panel*: numbers of models which converged to a solution within 1, 000 iterations during training (dashed lines) out of 100 models with different initial guesses. Also shown are the number of models that converged to a solution with self-transition probabilities higher than 0.8 (solid lines). **B:** Same as panel A for an SNN dataset comprising 9 neurons, 36 trials, and 14 s per trial. **C:** Same as panel A for an EXP dataset comprising 9 neurons, 36 trials, and 14 s per trial.

analogously modified by replacing the LL with the log-posterior, see Materials and Methods, Sec. "BIC and AIC").

In conclusion, selecting an appropriate model boils down to picking the optimal *m* value in the elbow region of the cross-validated LL. The examples of this section suggest using BIC to guide this choice. In Sec. "Model selection on ground truth datasets", we perform a more systematic analysis to validate this hypothesis in ground-truth datasets.

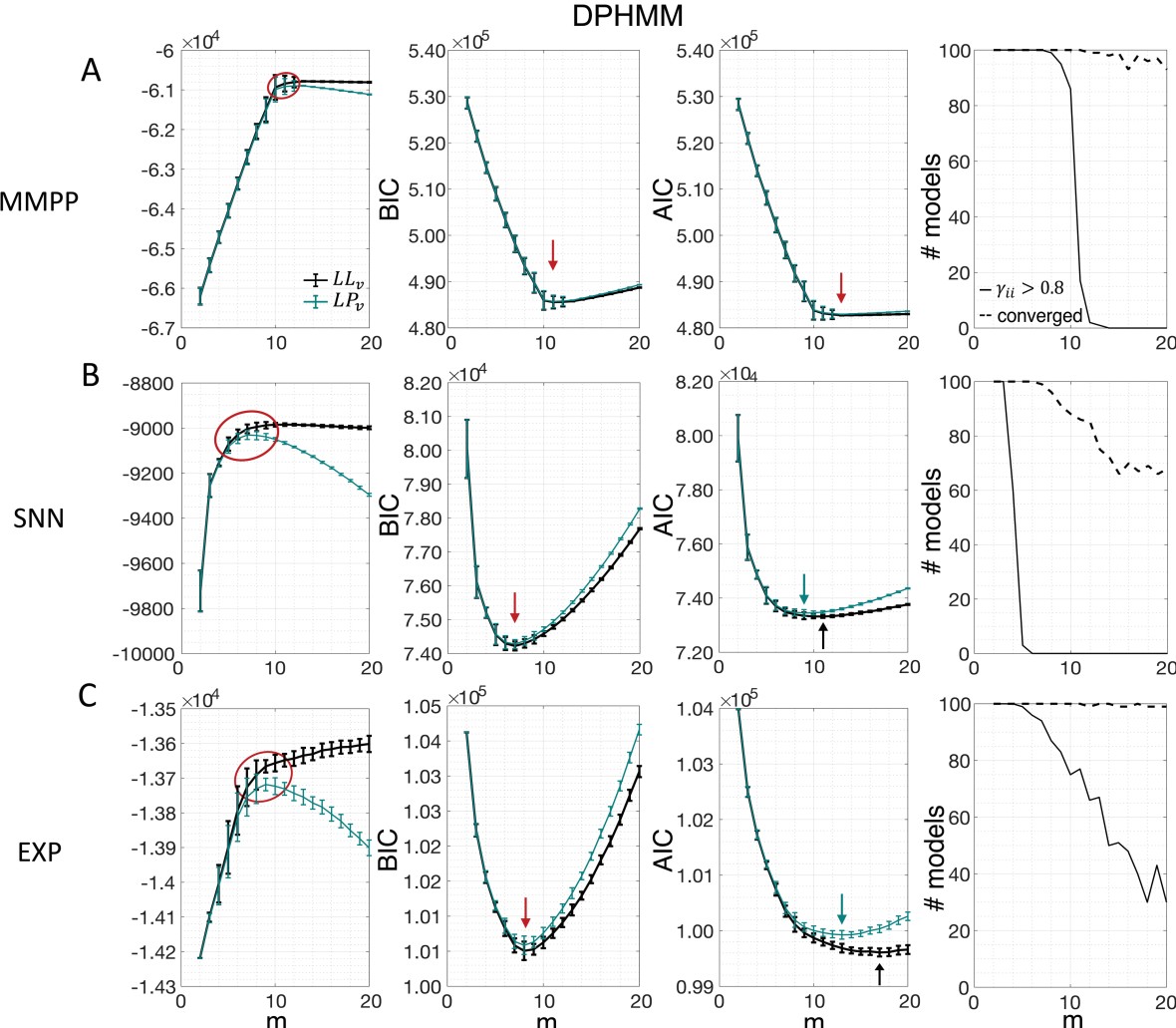

**Fig 4. Examples of model selection methods for the DPHMM.** *Black curves:* same as Fig 3 except these results were obtained with the DPHMM (same datasets as in Fig 3). *Green curves:* same as black curves but for the log-posterior probability $LP_v(m)$ and the modified BIC and AIC scores (see the text). Black arrows indicate the minimum of BIC or AIC when the log-likelihood is used; green arrows indicate minima when the log-posterior is used; and red arrows when the minima are the same for both.

## Temporal state persistence

The potential impact of low or intermediate self-transition probabilities is shown in Fig 6A-B for one cortical dataset. In this case, BIC gives $m^* = 8$ hidden states for the PHMM and $m^* = 7$ states for the DPHMM, compatible with the elbow region of $LL_v(m)$. However, in both the PHMM and DPHMM, all converged models with $m \geq 4$ had at least one state with self-transition probability lower than 0.8 (Fig 6A-B, right-most panels). The best models in this case exhibit very rapid state switching after decoding (Fig 6A-B, bottom panels). The sPHMM, on the other hand, only converged for models with $m \leq 3$ states (by construction, since this algorithm either converges to large self-transition probabilities, or doesn't converge at all). BIC gave $m^* = 3$ for the best sPHMM model, which exhibits no rapid state switching after decoding (Fig 6C, bottom panel). Visual inspection of the decoding shows that the sPHMM is

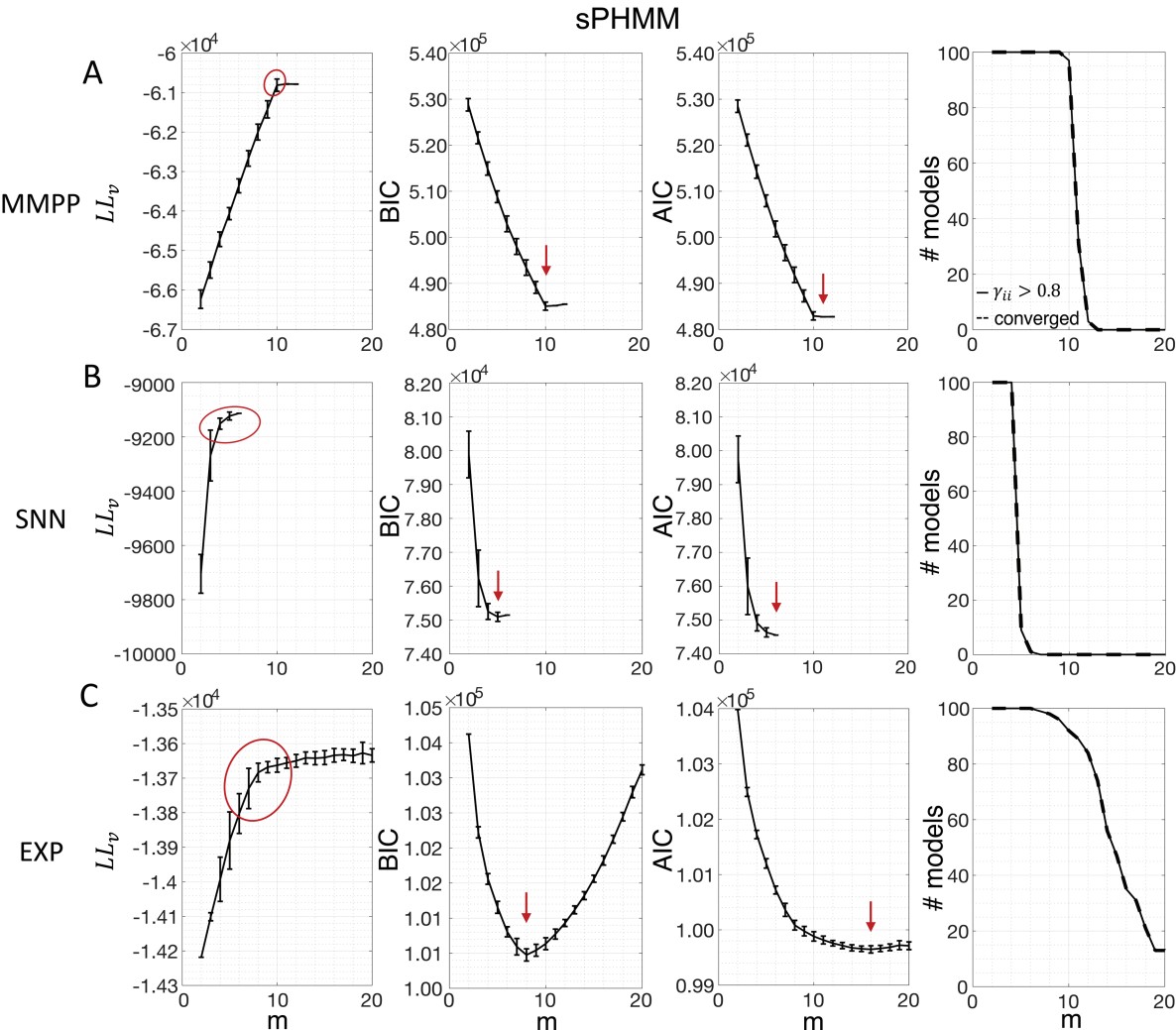

**Fig 5. Examples of model selection methods for the sPHMM.** Same as Fig 3 except these results were obtained with the sPHMM (same datasets as in Fig 3). Note that all models that converged (rightmost column) did so with $\gamma_{ii} > 0.8$.

the only sensible description of the data in this case, as the orange state seems to cover a long segment of the data where the neurons' firing rates do not appreciably change.

The value of the threshold, $\theta$, is directly related to the expected state durations and the bin size used to discretize time during training. Specifically, for a mean state duration $\langle T \rangle$ and a bin size $dt$, one should set a threshold $\theta = 1 - dt/\langle T \rangle$ (see S1 Appendix). A threshold of 0.8 corresponds to 5 time bins, or expected state durations larger than 100 to 500 ms for bins of 20 to 100 ms. This is in keeping with typical state durations in cortical datasets [10,13,15,17]. In some datasets, a lower threshold may be required. This is e.g. the case of our SNN datasets, where lower values for the self-transition probabilities are compatible with short transients generated by the simulation of the spiking network (not shown).

Since $\theta$ depends on the time bin $dt$, a different threshold should be used when using a different time bin. These parameters must vary within a reasonable range: $dt$ cannot be too large (otherwise some transitions may be lost) and $\theta$ cannot be too small (otherwise rapid state

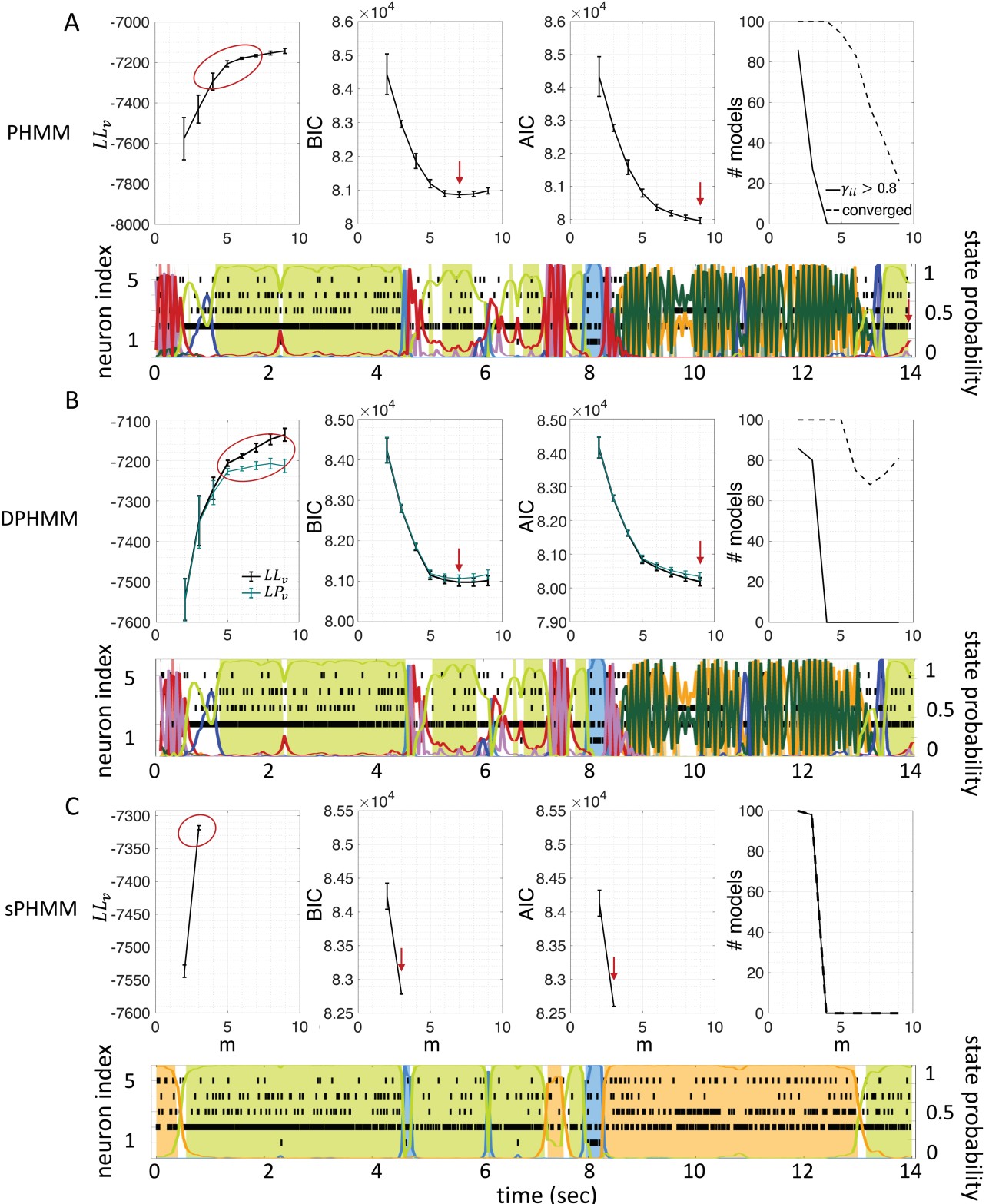

**Fig 6. Elimination of rapid state switching by sPHMM. A:** Top panels: same as Fig 3A for the PHMM trained on an EXP dataset. Bottom panel: rasterplot and PHMM decoding of one trial with $m^* = 8$ (minimum of BIC). **B:** Same as panel A for the DPHMM, with $m^* = 7$. **C:** Same as panel A for the sPHMM, with $m^* = 3$.

switching may result). Within this range, we found that the sPHMM algorithm is robust to changes in bin size and threshold corresponding to the same mean state duration (S1 Fig).

## Model selection on ground truth datasets

The results presented so far suggest that BIC performs well and reliably for model selection; in particular, BIC is in good agreement with cross-validation when the latter gives a clear answer. To assess the generality of these claims, we performed a systematic analysis of the sPHMM trained on MMPP datasets, for which ground truth is known. We chose the sPHMM because, among the three algorithms, the sPHMM is the most effective at preventing rapid state switching. The MMPP datasets were generated using random transition probability and firing rate matrices with 10 or 20 firing rates, each randomly chosen between zero and 30 spikes/s, and using either 100 or 1,000 random initial guesses for the model parameters (see Materials and Methods for full details). Fig 7A-D shows the optimal number of states inferred by cross-validation, BIC, and AIC compared with ground truth (black line). Three different ways were used to determine the optimal number of states under cross-validation: one based on the maximal value of the average $LL_v$ curve ($CV_{max}$); one based on the maximal change in slope of the average $LL_v$ curve ($CV_{slope}$); and one based on the one standard deviation method ($CV_{1SD}$; see below). Let $m_{max}$ be the value of $m$ for which $LL_v$ is maximal and $m_{slope}$ the value of $m$ for which $LL_v$ has the maximal change in slope. We found that $m_{max}$ tends to overestimate the true number of states, while $m_{slope}$ tends to underestimate it, often severely. AIC tends to overestimate the true number of states, especially when the latter is larger than 10. BIC tends to underestimate the number of states for 20 or more states in datasets with 10 neurons (Fig 7A-B), but it generally captures the ground truth in datasets with 20 neurons (Fig 7C-D). Similar results were obtained with 50 neurons (not shown). $CV_{1SD}$ selects the minimal value of $m$ for which one standard deviation (SD) above the average $LL_v(m)$ is larger than one SD below the average $LL_v(m_{max})$ [24,p. 244]. This method gives the most accurate cross-validation estimate of the true number of states and tends to agree well with BIC for 10 neurons. For 20 neurons, $CV_{1SD}$ tends to underestimate slightly the true number of states for $m > 15$ (in the experimental datasets, BIC and $CV_{1SD}$ tend to disagree more; not shown). Overall, BIC was the best model selection criterion among those tested.

As control, we performed the same analysis after shuffling the data [15–17,25] (Fig 7E-F). The comparison with shuffled datasets ensures that states and state transitions are true properties of the data, rather than the spurious result of training the model. We shuffled the data with two algorithms, named 'circular shuffle' and 'swap shuffle' [25]. A circular shuffle independently shifts each spike train by a random interval of time, while the swap shuffle randomly swaps the time bins (hence the spike count vectors). The circular shuffle preserves the firing rates of single spike trains, but impacts the correlations among the spike trains, so that the original state transitions are lost. The swap shuffle keeps the transitions intact, but it scrambles the hidden states into non-adjacent segments of shorter duration. Both procedures preserve the individual neurons' trial-averaged firing rates, but are expected to disrupt one or more aspects of HMM inference.

After training the sPHMM on circularly shuffled data, we computed the $LL_v$, BIC and AIC curves as done for the original MMPP data. These curves had the same general trend as the original data (not shown), but resulted in poor estimation of the number of hidden states (Fig 7E-F), although some patterns survived the randomization procedure (e.g., AIC and $CV_{max}$ tend to penalize too little while $CV_{slope}$ tends to penalize too much). The swap shuffle procedure had more dramatic effects, since the sPHMM did not converge on any of the 72 datasets during training.

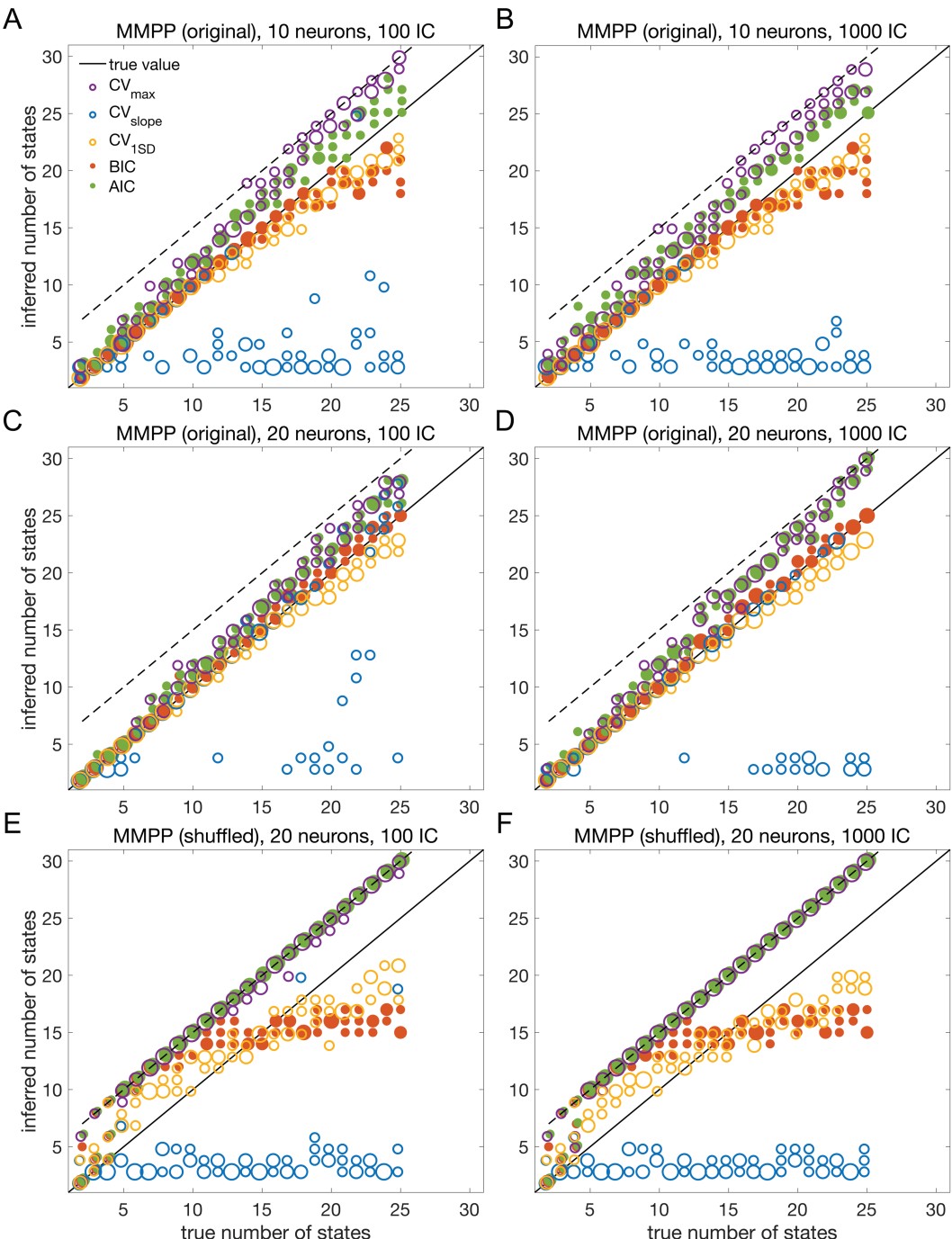

**Fig 7. Model selection performance of sPHMM in ground truth datasets. A:** Number of states inferred by sPHMM vs. the true number of states in MMPP datasets with 10 neurons and different numbers of hidden states (72 datasets in total, 3 datasets for each $m$) and 100 random initial guesses for the parameter values. The number of states used for HMM training ranged between 2 and the true number of states plus 5 (dashed line). Solid line: true number of states used to generate the MMPPs. Open circles: optimal number of states estimated by cross-validation (using three different methods; see the text). Red dots: optimal number of states estimated by BIC. Green dots: optimal number of states estimated by AIC. Some markers in the figure include more than one data point (reflected in the marker size). **B:** Same as panel A but with 1,000 random initial guesses. **C-D:** Same as panels A-B but for an MMPP with 20 neurons. **E-F:** Same as panels C-D for the circularly shuffled MMPP datasets, see the text for details.

The overall results for both the original and shuffled datasets were the same with either 100 or 1000 random initial guesses, an issue we consider in more detail later on.

## Application to experimental datasets

The analysis of the previous section indicates BIC as the best criterion for model selection, however the analysis was conducted on surrogate data (MMPP datasets). Do the conclusions hold for experimental datasets, for which there is no ground truth? In an attempt to answer this question, we repeated the analysis of Fig 7 on MMPP datasets that, rather than being randomly generated, were generated by HMMs fitted to EXP datasets (see Materials and Methods). These surrogate datasets have similar statistical properties as the experimental data, with the added benefit that ground truth is known. Fig 8 shows an example. The datasets used in this example were generated starting from the best sPHMM fits (for different $m$ between 2 and 8) to one EXP dataset; the corresponding $LL_v$, BIC and AIC curves are shown in Fig 8A. For each $m$, we used the best sPHMM to generate a surrogate (MMPP) dataset. A similar analysis as for Fig 7 was then conducted on the MMPP datasets, i.e., we trained new sPHMMs with varying $m$ on each surrogate dataset to infer the optimal number of states. The inferred number of states given by the different model selection methods are shown in Fig 8B. The results are in keeping with those of Fig 7A. BIC captured the ground truth when $m \leq 6$, and overall BIC and $CV_{1SD}$ performed best.

BIC eventually underestimates the true number of states because additional $\ln D$ terms in Eq 1 subdue the increase in LL, resulting in a higher BIC than achieved with a smaller number of states. This is more likely to occur the shallower the BIC curve is around its minimum (see middle panel of Fig 8A). The opposite occurs for AIC, where adding states causes a smaller additional penalty compared to the gain in LL.

## Full model comparison

So far we have used the number of states as a proxy for the ability of the selected model to provide a good description of the data. However, obtaining the correct number of states

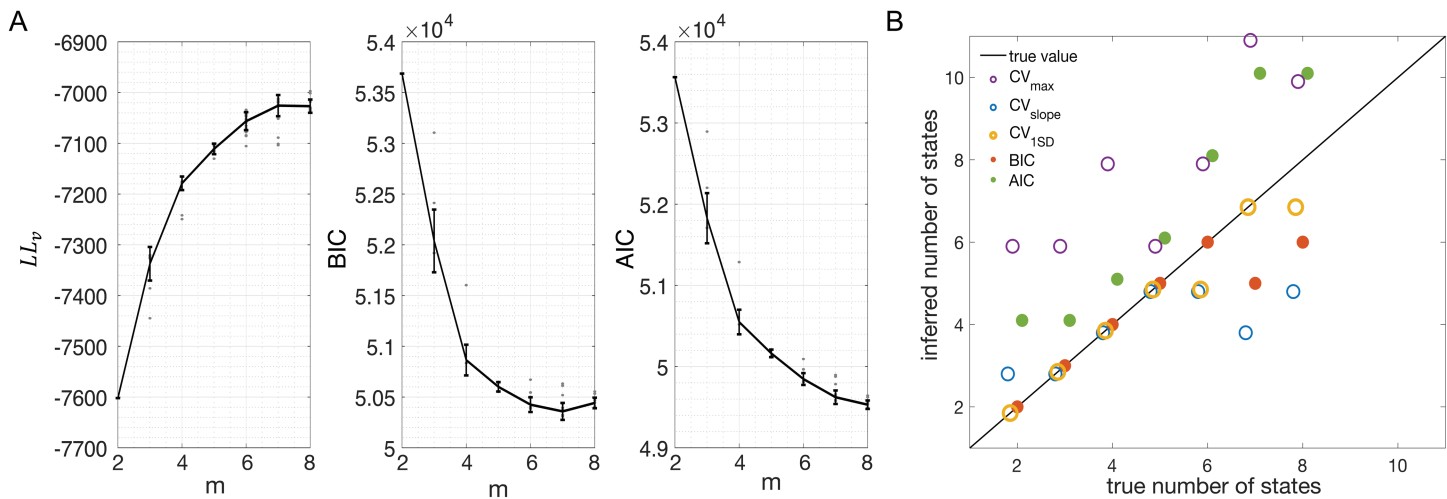

**Fig 8. Model selection on MMPP datasets obtained from fitting an sPHMM to an EXP dataset. A:** $LL_v$ curve (left), BIC score (middle) and AIC score (right) for the sPHMM trained on one EXP dataset using different values for $m$. Same keys as in Fig 3. **B:** Inferred numbers of states by sPHMM trained on MMPP datasets obtained from the best HMMs of the experimental datasets of panel A (see the text for details). Same keys as in Fig 7A.

(when ground truth is known) is no guarantee that the states identity and the state transitions are also correctly captured. To quantify how well those properties are captured by the selected model, we introduced a new index $\rho$ that measures the 'distance' between two models for a given dataset,

$$\rho(\Theta_1, \Theta_2) = \frac{D(\Theta_1)}{D(\Theta_2)},\tag{3}$$

where $\Theta_i = (\Lambda_i, \Gamma_i)$ are the parameters of trained HMM models that need comparison. $D(\Theta)$ was defined as the sum of the squared differences between the observed and inferred spike counts in each bin (see Materials and Methods for details). $D(\Theta)$ quantifies the irreducible variability of the data under the assumption that it was generated by an HMM with parameters $\Theta$. It does so by combining information about both the firing rate vectors defining the hidden states and the state transitions. If two models have the same states, the same state transitions, and the same transition times on a given dataset, we expect $\rho \approx 1$. When a true model is available, we set $\Theta_2 = \Theta_{true}$ and expect $\rho \geq 1$. In that case, $D(\Theta_{true})$ is computed after decoding the data with the model used to generate it (to account for the variability due to state transitions occurring inside a bin).

We show in Fig 9A the ratio $\rho(m) = \rho(\Theta(m), \Theta_{true})$ for an sPHMM (the 'test' model) trained on an MMPP dataset with 20 neurons and 10 states, for which the true model is available. The test model was trained with $m = 2, \dots, 12$ states (the model did not converge for $m > 12$). $\rho(m)$ decreases for $m \leq 10$ and reaches $\rho \approx 1$ for $m \geq 10$. $\rho(m) > 1$ for $m < 10$, indicating that the model $\Theta(m)$ did not capture the state transition times and/or the firing rates generated by the true model as well as the model with $m = 10$. For comparison, we show in the inset of the same figure the $\rho$ value for $m = 10$ obtained after randomly shuffling the states' firing rates and the off-diagonal transition probabilities. This is the value expected by chance when comparing the true model with a model with the same firing rates and transition rates as the test model with $m = 10$. Note that the $\rho(m)$ and $LL_v(m)$ values are highly correlated (Fig 9B), showing that $\rho$ and a distance measure based on the log-likelihood [28] are closely related. However, unlike the likelihood, our $\rho$ index plateaus at the correct number of states.

The values obtained for $\rho$ vary across a short range (from 1 to $\approx 1.2$), suggesting that $\rho$ values slightly larger than 1 already indicate a meaningful mismatch between models. To get a feeling for the meaning of the numerical $\rho$ values, we show in Fig 9C the decoding comparison between the true model and the models with $m = 5, 10$ and 12. The model with $m = 5$ gets most state transitions right but it mislabels some of the states (due to lack of states), and it gives $\rho \approx 1.1$. The model with $m = 10$ practically gives identical results to the true model and it gives $\rho = 1$. The overfit model with $m = 12$ is similarly good and also gives $\rho = 1$, but it is occasionally undecided on how to classify some bins (white space prior to the grey state). This phenomenon occurs due to the 'duplication' of states in an overfit model, as we show next (the following arguments apply in the absence of rapid state switching).

The reason for which the models with 11 and 12 states perform almost as well as the model with $m = 10$ is over-segmentation: the extra states 'duplicate' some of the original 10 states. We show this in Fig 10. Fig 10A shows the matrix of Euclidean distances between the firing rate vectors of the true model and the test model trained with $m = 12$ states. To obtain this matrix, we first matched the states of two different models according to their similarity using the Hungarian algorithm [29] (see Materials and Methods); the states were then relabeled so that the first 10 states in the two models are most similar to each other (blue diagonal) while

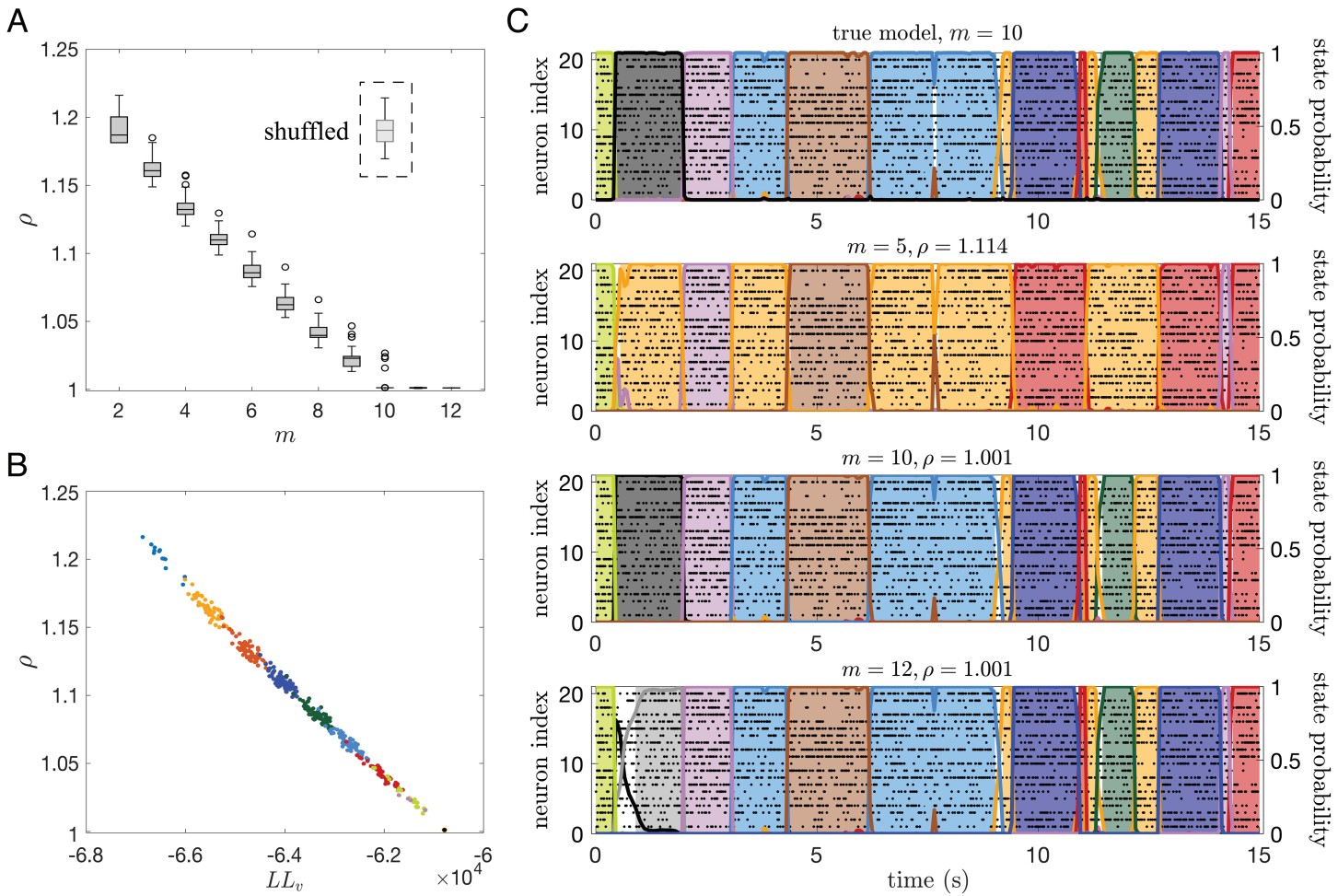

**Fig 9. The $\rho$ index for comparing HMMs. A:** $\rho(m) = \rho(\Theta(m), \Theta_{true})$ for an sPHMM with $m$ states ('test' model) trained on an MMPP dataset with 20 neurons and 10 states (50 trials, split into 40 and 10 in the training and validation sets, respectively). $\rho$ decreases with $m$ until $m$ reaches the true number of states $m^* = 10$. For each $m$, 100 HMMs were trained with different random initial guesses for the model parameters. No HMM model converged when $m > 12$. *Dashed box:* the $\rho$ value for $m = 10$ after randomly shuffling the states' firing rates and the off-diagonal transition probabilities. **B:** Log-likelihood of the test model on validation sets increases as a function of $m$ and is highly anti-correlated with $\rho(m)$. Different colors correspond to different values of $m$. **C:** Examples of sPHMM's posterior decoding of the best test models (i.e., having the largest likelihood on the training set) trained with 5, 10, and 12 states on the MMPP dataset of panel A, compared with the true model (having 10 states; top panel).

different states are less similar. The additional states (11 and 12) of the test model are very similar to states 10 and 5 of the true model, respectively.

Fig 10B shows the comparison between the states of the two models: the firing rate vectors of the true model (black rectangles), the firing rate vectors of the best-matching 10 states in the test model (blue filled rectangles), and the firing rate vectors of states 11 and 12 of the test model (red filled rectangles), the latter superimposed, respectively, onto states 10 and 5 (which are most similar to them). The true states are estimated extremely well by the trained HMM, except for states 5 and 10, which are estimated less accurately, with two states used for each. This, however, results in similar $\rho \approx 1$ values (Fig 9A).

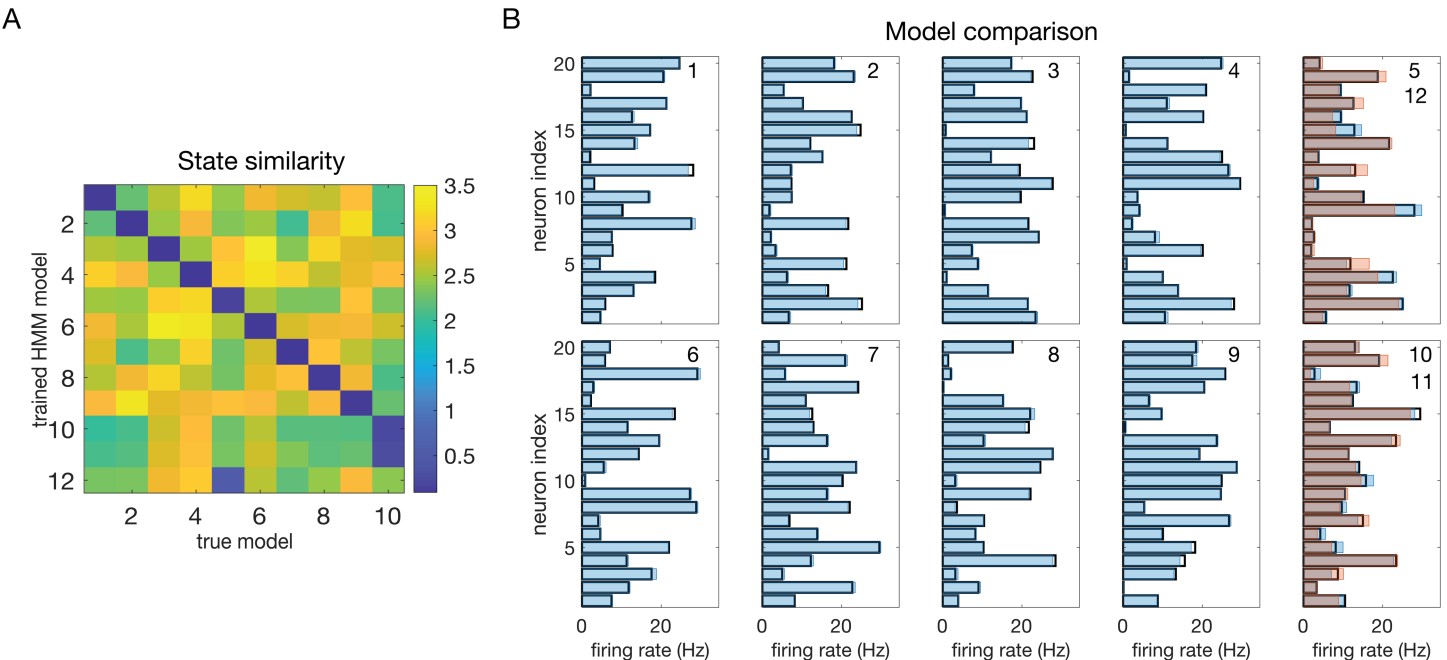

**Fig 10. Comparison of HMMs.** States' comparison between the trained HMM with 12 states and true model (same models and dataset used in Fig. 9). **A:** Similarity between the true states and the states of the test model. The entry in row $i$ and column $j$ is the Euclidean distance between the firing rate vector of state $i$ of the trained model and the firing rate model of state $j$ of the true model, with the states' labels in the two models matched for maximum similarity (see the text). **B:** Each panel shows the firing rate vectors of the true model (black rectangles) and the firing rate vectors of the best-matching 10 states in the trained HMM model (blue filled rectangles). For states 5 and 10, the firing rate vectors of states 11 and 12 of the test model are also shown (red filled rectangles).

## The role of the initial parameter guesses

The HMMs were fit to the data by the method of expectation-maximization [30,31]. When this method converges, it is only guaranteed to converge to a local maximum of the log-likelihood. One way to avoid getting stuck in a local maximum is to run the algorithm for many initial parameter guesses, and then select the model giving the best results. In most of the results shown so far we have used 100 random initial guesses for each $m$, and have selected the model based on the best average performance across all initial guesses. However, we must tune $m(m-1) + mN$ parameters during training (where $m$ is the number of states and $N$ is the number of neurons, see Materials and Methods). For 10 states and 10 neurons this is about 200 parameters, presumably giving rise to a rugged LL landscape with many local maxima and saddle points. Larger datasets make this situation even worse, suggesting that 100 random initial guesses should be an insufficient number. However, Fig 7 shows similar results for 100 vs. 1,000 random initial guesses. This weak dependence on the number of initial guesses was confirmed in the experimental datasets, where we performed the more systematic analysis shown in Fig 11. As we have no ground truth for EXP datasets, we took the number of states inferred using 1,000 initial guesses as a proxy for the true value (called $M^*$). We found that for 100 or more random initial guesses, $m^*$ differed from $M^*$ in 20% or less of the cases, with a mismatch of only one unit (a mismatch of two units occasionally occurred with 10 random initial guesses).

To verify whether using the same number of hidden states also leads to similar states and state transitions, we repeated the analysis of the previous section using the best model – trained with 1,000 random initial guesses – as a proxy for the true model. As test models,

hidden states vs. initial conditions (EXP)

**Fig 11. The role of random initial guesses.** Impact of random initial guesses on the HMM analysis of the experimental datasets (EXP). *Top:* histogram of mismatches between $m^*$, the inferred number of states, and $M^*$, the number of hidden states inferred when using 1,000 random initial guesses (used as proxy for the true number of states). Each bar is the average across 100 sets of random initial guesses, as a function of the number of initial guesses. *Bottom:* dot plot showing the results for all datasets. Each column of dots (within a box) represents a dataset. Within each column, each dot represents the outcome obtained with one set of random initial guesses. A dot in the upper band is an outcome with $m^* = M^* + 1$; a dot in the next band below is an outcome where the inferred $m^*$ matched $M^*$; a dot in the band further below represents a case where $m^*$ was 1 less than $M^*$; and so on.

we used models trained with 100 or 1,000 random initial guesses. In both cases, test models with the same number of states as the 'true' model gave a $\rho$ value close to 1 (S2 Fig, blue histograms). These results show that using from 100 or 1,000 initial guesses results in the same states, state transitions and transition times as for the best model with 1,000 initial guesses, as long as the number of hidden states are the same.

For comparison, randomly shuffling the states' firing rates and the off-diagonal transition probabilities gives a higher value of $\rho \approx 1.2 \pm 0.1$ (S2 Fig, red histograms). Such a $\rho$ value, as we know from Fig 9, gives a poor match to the true model. The probability of getting the trained $\rho$ values under the shuffled model was 0.0098 or smaller (S2 Fig, area of the red histograms extending up to the rightmost bin of the blue histograms), revealing that is highly unlikely to obtain $\rho \approx 1$ by chance from a non-matching model.

These results show that, perhaps surprisingly, a relatively small number of initial guesses gives results comparable to those obtained with a much larger number. There are, in principle, better ways to assign the initial parameter values, rather than using random ones. We attempted to infer the initial firing rate vectors from the data by using clustering algorithms (see Materials and Methods), however we found no discernible improvement (see Discussion).

## Discussion

In this work we have analyzed the problem of fitting Poisson-HMMs to ensembles of spike trains, and have addressed two main problems that often occur: (i) cross-validation typically does not provide a clear log-likelihood maximum to determine the optimal number of states, and (ii) HMMs can produce rapid state switching that is incompatible with the data. We have

found that the first problem can be at least partially solved by using BIC for model selection, or by cross-validating the log-posterior probability when training the HMM with a prior on the transition probabilities. The second problem can be solved by using a 'sticky' algorithm for fitting a Poisson-HMM. Combining training using the sticky algorithm and BIC for model selection successfully prevents unreasonably frequent state transitions and leads to optimal inference in surrogate data built to resemble the neurophysiological data.

We have also studied the role of random initial guesses for the model parameters and found that the optimal number of states can typically be obtained starting from a limited pool of random initial guesses ($\leq 100$). Although this is possibly helped, in the sPHMM, by the fact that this algorithm changes the initial guesses 'on the fly' when a bad solution is found, analogous results found with the other algorithms suggest that this result is more generic and applies to non-sticky algorithms as well. Overall, our results show that the sPHMM method is an efficient and reliable method to fit a Poisson-HMM to simultaneous recordings of multiple spike trains.

We discuss below in some detail the problem of model selection, the phenomenon of rapid state switching, and the dependence on the random initial guesses. The underlying inference problems are always stated in terms of correctly inferring the number of hidden states, assuming that two HMMs with the same number of states will provide equivalent models. We have shown that this is the case by using a new index $\rho$ that quantifies the triple match between states' firing rate vectors, state transitions, and state transition times during decoding. The index is close to 1 for two models for which these features match with high accuracy. When ground truth is available, $\rho(m) \geq 1$ and we found that the correct number of hidden states is the smallest $m$ for which $\rho(m) \approx 1$. Using this method, we have also found that models having the same number of states but starting from different initial parameter guesses converge to equivalent models.

## The problem of model selection

We have compared model selection results obtained with cross-validation, BIC and AIC. Such methods are widely used and all attempt to reduce the risk of overfitting. Overfitting occurs when the model has good performance on the training set but poor performance on data not used for training. Although they have different origin and motivation, BIC and AIC can be understood (with different caveats) as methods to estimate the in-sample error [24]. In turn, the in-sample error can be used as a proxy for the average out-of-sample error, i.e., the performance of the model on test data not used for training. How successfully BIC and AIC achieve this goal depends on many factors including the model class being tested, the data generating mechanism, and the size of the dataset. Cross-validation is a method to estimate directly the out-of-sample error and can be applied more widely [24]. Cross-validation, however, requires large enough datasets and it is computationally expensive, which is why BIC and AIC are often used instead. BIC and AIC have been frequently used in the HMM analysis of spike data [12,15,17,27,32–34].

Another motivation for using BIC or AIC rather than cross-validation is the fact that, in an HMM, the cross-validated likelihood tends to monotonically increase with the number of hidden states. The main reason is over-segmentation: when imposing a larger number of states than necessary, one state can be decomposed into multiple states, giving the model the same or larger likelihood on the validation set. As a consequence, the LL will not decrease as a function of $m$. These states tend to appear in sequence, either because they are 'copies' of the same true state, or because they lead to rapid state switching. With spike data, over-segmentation may further be encouraged by the need of inferring firing rates from spikes.

Because of the large variability of cortical spike trains, vectors of similar firing rates are difficult to separate.

Various authors have resorted to different strategies to overcome this problem. Some authors rely on a subjective estimate of the optimal number of states as the largest value of $m$ prior to a visible plateau in $LL_v(m)$ [16]. This approach has an element of arbitrariness as it depends on the subjective evaluation of the beginning of the plateau. Other authors have preferred to select the number of states *a priori* [6–10,35], especially when the data or the task structure suggest what the number of states should be. For example, data with ON-OFF states might suggest the presence of only 2 hidden states. This approach does away with model selection but should be avoided, unless one has very strong prior knowledge on the number of states. A method that seems to give good results in curtailing overfitting is to select the number of states that maximizes the LL during training, and then remove all states that never last longer than 50 ms after decoding [13,36]. This method has also been used in combination with BIC [15,17,34]; however, there is not much use for this method in the case of Poisson-HMMs, where bin sizes are larger and often lasting 50 ms or longer [25]. In the sPHMM, the appearance of very short-lived states during decoding is curtailed by imposing high self-transition probabilities.

In this work, we have found that BIC typically infers the true number of states when ground truth is available. The true number always falls within the elbow region of the cross-validated $LL_v(m)$ curve; however, algorithmic attempts to determine the optimal point directly from the $LL_v(m)$ curve did a poor job in our hands (the best method, $CV_{1SD}$, can differ substantially from BIC on experimental datasets, and is less reliable). Regarding AIC, we found that it tends to overestimate the true value of $m$, presumably due to a smaller penalization of the LL in large datasets compared to BIC. This is in keeping with the known empirical fact that AIC tends to perform better than BIC for small datasets, where BIC tends to penalize too much, while BIC often performs better than AIC for sizable datasets (like ours), where AIC tends to over-penalize.

In addition to BIC and AIC, there are many other information criteria available in the literature [37,38]. These include, for example, the 'generalized information criterion' (GIC), the 'corrected AIC' (AIC$_c$), and the minimum description length (MDL). These and many other information criteria take the general form $-2LL_t + \eta(K, D) \cdot K$, where $LL_t$ is the log-likelihood of the model evaluated on the training set, $D$ is the number of data points, and $K$ is the number of parameters (more precisely, $K$ is an appropriate measure of model complexity). However, the precise function $\eta$ is difficult to estimate, because it also depends on the model that generates the data. For AIC and BIC, $\eta = 2$ and $\eta = \ln D$, respectively. The AIC$_c$, which is the AIC corrected for small samples and is exact for linear regression models, has $\eta = 2D/(D - K - 1)$. Note that in this case $\eta \to 2$ for large datasets (retrieving the AIC), but it is $\eta > 2$ for finite datasets, giving a larger penalization compared to AIC. For our datasets, AIC$_c \approx$ AIC (for example: $D \sim 15,000$ and $K \sim 250$ for $m = 10$ states considering 50 trials of 15 s duration, divided into 50 ms bins). MDL is a criterion that somewhat interpolates between AIC and BIC, but it requires in each case a definition of description length which depends on the model under consideration; for regular parametric models, it tends to produce the same results as BIC [39]. We have focused on BIC and AIC for several reasons, including popularity and previous use in the HMM literature. Other methods, such as AIC$_c$ or MDL, would give similar results on our datasets.

When performing the max *a posteriori* estimate of the parameters for the DPHMM, we have adapted BIC and AIC by replacing the LL with the log-posterior. This is a pragmatical procedure for which we cannot provide a clear theoretical justification, especially for AIC (see Materials and Methods and below); in any case, the modified criteria led to the same

results for BIC, and while the modified AIC tends to reduce overfitting, it still favored models that are too complex. This does not detract from the merits of the DPHMM, which reside in encouraging models which reduce rapid state switching, and in providing a clear maximum of the cross-validated log-posterior for model selection.

Finally, we must mention that neither BIC nor AIC, and similarly other information criteria that are derived under the assumption of model regularity, can be theoretically justified for selecting the number of hidden states of an HMM. The reason is that an HMM is a singular model [40]. In a singular model, some manipulations that are required to derive BIC and AIC are not allowed. Alternative criteria have been developed for singular statistical models, known as "widely applicable information criteria" [41,42]. Those are similar to AIC or BIC, but differ in the exact form of the penalization term, which is however difficult to estimate for many models including HMMs. Instead, BIC and AIC are easy to use and can have good empirical success despite the fact that some of the assumptions made when deriving them do not hold for HMMs. This has also been our observation on our spike datasets, especially for BIC.

### The problem of rapid state switching

The problem of extremely rapid state switching (Fig 6) is due to intermediate-to-low self-transition probabilities, which in turn can be facilitated by over-segmentation due to over-fitting: when the number of states increases, the number of models with large self-transition probabilities decreases (Fig 3-5, rightmost plot of each panel). The sPHMM is effective at preventing over-segmentation by imposing that all self-transition probabilities are above a threshold. When such a solution exists, it may also depend on the initial parameter values whether an algorithm would converge to it. The sPHMM has a better chance at finding such solutions because it changes the initial guesses on the fly. Whenever the sPHMM training algorithm converges to a solution with low self-transition probabilities, it restarts from a better set of parameter values and imposes above-threshold values for the self-transition probabilities – thus enforcing large diagonal values as final conditions.

We caution that intermediate or even low self-transition probabilities do not necessarily lead to rapid state switching, but could capture instead the presence of many short-lived states. This situation can easily be reproduced in MMPP and SNN datasets (not shown). In such cases, if overfitting occurs without rapid state switching, a different problem may occur, namely, a larger fraction of undecided time bins during decoding (i.e., data bins not assigned to any state). Many undecided bins are likely the signature of over-segmentation, i.e., of the presence of spurious states with similar transition rates, preventing the posterior state probabilities from reaching the 0.8 threshold needed for decoding. Also in this case, the sPHMM offers a practical solution: by leading to a smaller number of hidden states, it will reduce over-segmentation and therefore the number of undecided bins. If this leads to underfitting of datasets with short-lived states, the solution is simply to lower the threshold value for the self-transition probabilities (see S1 Appendix).

### Dependence on the initial guesses of the model parameters

After a successful training of our HMMs, we are only guaranteed to have converged to a local maximum (or saddle) of the LL. This is a common problem in maximizing a likelihood function that is not convex. One way to circumvent this problem is to fit the same model starting from different random initial guesses of the model parameters, and then select the model with the largest LL. We found decent results with tens of initial guesses, and not much benefit in using more than 100. With more than 10 initial guesses, the mismatch with number of hidden

states inferred by using 1000 random guesses is at most one (Fig 11). These results somewhat justify the use of a small number of initial guesses seen in previous works, sometimes as small as 5 or 10 [12,13,15,17,32,35].

There are, in principle, better ways to infer the initial parameter values than using random ones. In Poisson HMMs, we used $K$-means and DBSCAN [43] to cluster the firing rate vectors across neurons, and used the centroids of the clusters as the rows of the initial firing rate matrix (see Materials and Methods). We did not get better results compared to using random initial guesses. For example, we compared 10 sets of initial guesses found via $K$-means (using each time random initial assignments of the centroids) vs. 100 random initial guesses, and the latter could lead to models with larger LLs (not shown). The problem may be due to the requirements of these alternative methods [44]. For example, we noticed that the true states in the MMPP datasets tend to occupy the edges of the clusters when using DBSCAN, whereas the cluster centers are then used as initial guesses. Obtaining good spherical clusters requires hand-tuning of parameters for different $m$ and for different datasets, which is hard. $K$-means incurs different problems: although it relies on a single parameter (the number of states, $m$), different runs of $K$-means starting from different initial guesses can give rather different results, so that, similarly to using random initial guesses, there is the issue of deciding how many initial guesses to use for $K$-means. In conclusion, while the search for a more principled method is worth pursuing, we have found that using 100 random initial guesses is straightforward and can outperform alternative methods based on clustering.

## Comparison among the algorithms

By imposing a Dirichlet prior on the transition probabilities, the DPHMM is a principled way to prevent rapid state switching and we have found that it works rather well. The sPHMM could be interpreted as a PHMM with a prior over the self-transition probabilities that is zero below 0.8 and uniform above it, with the prior being strictly enforced: the sPHMM will either converge to a model with large self-transition probabilities, or will not converge at all. In the latter case, an HMM is very likely an inadequate model to describe the data. Although the sPHMM and the DPHMM perform comparably, the sPHMM is more effective at preventing rapid state switching (Fig 6). Both algorithms are preferable to the standard PHMM.

The MHMM tends to have a reliable performance and is less frequently affected by the problem of rapid state switching than the PHMM. However, this model requires a very small time bin (which, on the plus side, allows for a high temporal resolution of the state transition times). The requirement for short bins is due to the assumption that at most one neuron can emit a spike in each bin. Legitimate bin lengths will depend on the firing rates of the neurons. With bins widths longer than 10 ms, the MHMM model generally cannot be used, because multiple spikes from each neuron are expected and multiple neurons will likely emit spikes in the same time bin. This problem is especially severe in datasets with large numbers of neurons [45]. For example, for an ensemble of 100 neurons (assumed conditionally independent given the state) having common firing rate as low as 5 spike/s, and time bins of 2 ms, there is a 0.26 probability that multiple neurons will fire a spike in the same bin. From this point of view, Poisson HMMs are a good solution, because they require larger bin widths than MHMM, but not too large. Time bins of 20-100 ms have been used in the hippocampus [25], and bin widths as short as 10 ms have been used in pre-motor and gustatory cortices [11,46]. Bin widths of this length are pretty short and allow to estimate state transition times rather accurately.

## Alternative approaches

Generalizations of the HMMs studied here are possible in several directions, and include combinations with generalized linear models to account for non-stationarity [47,48], hidden semi-Markov models to account for non-exponential distributions of state durations [49,50], and Bayesian non-parametric HMMs which do not require separate model selection [5,26,51,52]. In particular, the infinite-HMM [53] extends hidden Markov models to have a countably infinite number of hidden states. In this approach, one tries to learn a few hyperparameters defining a hierarchical Dirichlet process that controls the 'effective' state-transition matrix and the expected number of distinct hidden states in a finite sequence. This approach had been used for spike data [52], fMRI data [5], and more recently for animal behavior [54]; its motivation is to have a more flexible model for complex data structures and to avoid the limitations of model selection for finite HMMs. However, inference algorithms for Bayesian nonparametric models are computationally demanding and can be prohibitive for multi-dimensional data or for large datasets; moreover, they are more involved than their parametric counterparts, typically requiring variational approaches or Markov Chain Monte Carlo methods [51]. Finally, the infinite-HMM does not prevent rapid state switching [26]. Our motivation was to provide a heuristic solution to the related problems of model selection and rapid state switching that requires no changes to the well established Baum-Welch algorithm: our sPHMM just avoids rapid state switching by selecting new initial conditions at certain points during training. It is fast, easy to implement and always provides a solution when the algorithm converges. Moreover, it provides a conservative number of hidden states, reducing the chance of overfitting. These properties make it an ideal practical solution to the use of HMM in biology and neuroscience.

## Conclusion

We have analyzed in detail the problem of fitting HMMs to spike data, focusing on two main issues involving model selection (monotonic $LL_v(m)$ curve) and the temporal persistence of hidden states (rapid state switching). Regarding the first issue, we have shown that BIC allows to capture ground truth in surrogate data, while agreeing with the elbow region of the cross-validated LL curve in the experimental datasets. Regarding the issue of rapid state switching, we have presented two 'sticky' versions of Poisson-HMM, one with a Dirichlet prior over the transition probabilities (DPHMM), the other enforcing self-transition probabilities to be above a threshold (sPHMM). Both algorithms are effective at reducing or, in the case of the sPHMM, eliminating extremely rapid state switching. The sPHMM is an easily implementable and reliable algorithm, and via the accompanying code, readily available to the neuroscience community.

## Materials and methods

### Datasets

The analyses reported in this article were performed on 3 different types of spike data, described below.

**Electrophysiology data (EXP).** The experimental data were sessions of simultaneously recorded neurons from the gustatory cortex (GC) of behaving rats first reported in [23]. Briefly, movable bundles of 16 microwires were implanted bilaterally in the gustatory cortex and intraoral cannulae were inserted bilaterally. After postsurgical recovery, rats were trained to self-administer fluid tastants through intraoral cannulae by pressing a lever under head-restraint within 3 s presentation of an auditory cue (a 75-dB pure tone at a frequency of

5 kHz). These are dubbed 'expected trials'. In some of the trials, additional tastants were automatically delivered through the intraoral cannulae at random times near the middle of the intertial interval and in the absence of the anticipatory cue ('unexpected trials'). The following tastants were used: 100 mM NaCl, 100 mM sucrose, 100 mM citric acid, and 1 mM quinine HCl. Five seconds after the delivery of each tastant, water (50 $\mu l$) was delivered to rinse the mouth clean through a second intraoral cannula. During each session, multiple single-unit action potentials were amplified, bandpass filtered, and digitally recorded. Single units from 37 sessions were isolated using a template algorithm, clustering techniques, and examination of interspike interval plots (Offline Sorter, Plexon). Each session comprised around 36 trials, each lasting 14s, starting 8s prior to the occurrence of a stimulus and ending 6s after stimulus delivery. Starting from a pool of 299 single neurons in 37 sessions, neurons with peak firing rate lower than 1 spike/s (defined as silent) were excluded from further analysis, as well as neurons with a large peak around the 6–10Hz in the power spectrum of their spike trains, as such neurons were classified as somatosensory (i.e., modulated by mouth movements [55,56]). Only ensembles with five or more simultaneously recorded neurons were included in the analyses reported here.

Datasets from other brain areas were also used, specifically: medial prefrontal cortex [56], orbitofrontal cortex [23], basolateral amygdala [23], and gustatory thalamus [57]. The HMM analysis of these datasets shared similar features and led to similar conclusions as for the main dataset.

**Markov-modulated Poisson process (MMPP).** The second type of data was generated by computer simulations of a continuous-time Markov process with $m$ states and transition probability matrix $\Gamma$, emitting ensembles of Poisson spike trains with a given firing rates matrix $\Lambda$. While in state $i$, Poisson spike trains with rates $\lambda_{ni}$ were generated by sampling their interspike intervals $\Delta t$ from the exponential distributions $\rho(\Delta t) = \lambda_{ni} e^{-\lambda_{ni}\Delta t}$, where $n = 1, ..., N$ is the neuron index. The state sequences occurred according to the transition matrix $\Gamma$ and were generated using the Gillespie algorithm [58,59]. In the literature, such Markov model has been called a 'multivariate Markov-modulated Poisson process' (MMPP) (e.g. [60]). The advantage of using an MMPP is that everything is known about the process generating the data, and the Poisson-HMM is a true model for it.

We built the MMPP by using two different procedures, leading to two types of datasets. In the first case, $\Gamma$ was randomly generated with diagonal elements uniformly distributed between 0.8 and 1, while the firing rates were uniformly sampled between 0 and 30 Hz across 10 or 20 neurons and $m$ states, with $m$ ranging from 2 to 25. This method was used in Figs 3–5 and 7.

The second type of MMPP aimed to mimic electrophysiological data (Fig 8). For a given session in EXP, we trained the sPHMM for $m$ values ranging from 2 to 15 using 100 random initial parameter guesses for each $m$ (for $m > 8$ the training algorithm did not converge within 1,000 iterations). For each $m$, we selected the HMM with maximum likelihood on the training set, providing a Markov model with $m$ states, a transition probability matrix $\Gamma$, and a firing rates matrix $\Lambda$. Such a model was then used to generate the MMPP.

For the first type of MMPP, we generated datasets with 50 trials, each lasting 14 seconds; for the second type, the number of trials and the duration of each trial were the same as in the EXP dataset used to infer the MMPP.

**Spiking Neural Network (SNN).** The third dataset was generated by computer simulations of a spiking network with a clustered architecture [13]. The network had 4000 excitatory ($E$) and 1000 inhibitory ($I$) leaky integrate-and-fire (LIF) neurons. Neurons were randomly connected with probabilities $p_{EE} = 0.2$, $p_{EI} = 0.5$, $p_{IE} = 0.5$ and $p_{II} = 0.5$, where $p_{ab}$ is the probability that a presynaptic neuron of type $b \in \{E, I\}$ is connected to a postsynaptic neuron of

type $a \in \{E, I\}$. The synaptic weights between presynaptic neurons $j$ and postsynaptic neuron $i$, $w_{ij}$, were drawn from a Gaussian distribution with means $w_{EE} = 0.0416$, $w_{IE} = 0.0212$, $w_{EI} = -0.0882$, and $w_{II} = -0.0848$ and standard deviations 0.0028 ($w_{EE}$ and $w_{EI}$) and 0.0020 ($w_{IE}$ and $w_{II}$).

A subset of 3600 excitatory neurons were further partitioned into $Q = 10$ equal clusters of 360 neurons each, with the synaptic weights inside each cluster having a larger mean $J_+ w_{EE}$, with $J_+ = 2.2$, whereas the mean excitatory weights between $E$ neurons belonging to different clusters were set to a smaller value $J_- w_{EE}$, with $J_- = \max\{1 - \gamma(J_+ - 1)f/Q, 0\}$, where $\gamma = 0.5$ and $f = 0.9$ is the fraction of excitatory neurons organized in clusters.

The LIF neurons emitted a spike as soon as their membrane potential $V$ crossed a threshold $V_{th}$ ($-55$ mV for $E$ neurons and $-58$ mV for $I$ neurons), after which they were clamped to a value $V_r = -60$ mV for a refractory period of 5 ms.

The subthreshold dynamics of neuron $i$ of type $a \in \{E, I\}$ obeyed the equation

$$\frac{dV_i}{dt} = -\frac{V_i - V_L}{\tau_a} + \frac{I_i^{rec}}{C} + \frac{I_a^{ext}}{C}, \tag{4}$$

where $\tau_a$ is the membrane time constant ($\tau_E = 20$ and $\tau_I = 10$ ms), $V_L = -60$ mV is the resting potential, $C = 1$ nF is the membrane capacitance, $I_i^{rec}$ is the recurrent synaptic input to neuron $i$, and $I_a^{ext}$ is a constant external current (the same for all neurons of the same type: $I_E^{ext} = 0.3217$ nA and $I_I^{ext} = 0.2080$ nA).

The recurrent input current to neuron $i$, $I_i^{rec}$, was the sum of excitatory and inhibitory inputs, $I_i^{rec} = I_{iE}^{rec} + I_{iI}^{rec}$, with

$$\tau_{syn,b} \frac{dI_{ib}^{rec}}{dt} = -I_{ib}^{rec} + \sum_{j \in b} w_{ij} \sum_k \delta(t - t_k^{(j)}), \tag{5}$$

where $b \in \{E, I\}$ is the type of presynaptic neuron, $\tau_{syn,b}$ is the synaptic time constant ($\tau_{syn,E} = 8$ ms and $\tau_{syn,I} = 5$ ms), the first sum on the right hand side is over all presynaptic neurons of type $b$ connected to $i$, and the second sum is over all presynaptic spikes, $t_k^{(j)}$, emitted by presynaptic neuron $j$. According to this model, each presynaptic spike produces an exponential postsynaptic current, with decay time constant $\tau_{syn,E}$ for excitatory inputs and $\tau_{syn,I}$ for inhibitory neurons.

The network was numerically simulated using the Euler algorithm with a time bin of 0.005 ms for 2,000 seconds, and produced spontaneous (i.e., not stimulus-evoked) metastable dynamics. The whole simulation was then divided into non-overlapping trials of similar number and duration as the trials of the EXP datasets. The collections of such trials comprised one dataset. From each dataset, 9 neurons randomly chosen from 5 clusters were used to produce a single SNN dataset used for the HMM analysis.

## Hidden Markov Models

An HMM is a stochastic process characterized by $m$ (hidden) states, a matrix $\Gamma$ of transition probabilities $\gamma_{ij}$ from state $i$ to state $j$, and state-dependent probability laws $e_i(O)$. The latter is the probability of observation $O$ when in state $i$. Both $\gamma_{ij}$ and $e_i(O)$ are conditional probabilities: $\gamma_{ij} = P(j|i)$ and $e_i(O) = P(O|i)$, with $\sum_{j=1}^{m} \gamma_{ij} = 1$ and $\sum_O e_i(O) = 1$ (normalization). The initial conditions are specified by probabilities $\pi_i$ of being in state $i$ at the initial time. To train the model, time was discretized into bins of length $\Delta t$, with observations in each bin being vectors of spike counts across $N$ simultaneously recorded neurons (Poisson HMMs), or

a discrete set of $N + 1$ symbols (MHMM). Both the spike counts and the discrete symbols are to be interpreted as noisy observations of the hidden process generating the spiking activity of the neurons, assumed in all cases to fire spikes according to a Poisson process. This choice was motivated by technical convenience and the theoretical importance held by the Poisson process in modeling cortical spike trains. In all models, the hidden states could be univocally characterized by vectors of firing rates across the $N$ neurons. These firing rates were collected in an $N \times m$ matrix $\Lambda$, whose entries $\lambda_{ni}$ were the firing rates of neuron $n$ in state $i$. $\Lambda$ also completely characterizes the emission probabilities.

**Poisson-HMMs.** The following applies equally to the PHMM, the DPHMM and the sPHMM, collectively referred to as Poisson HMMs. The observations were vectors of spikes counts in time bins of length $\Delta t = 50$ ms, although bin sizes between 20 and 100 ms gave similar results. The probability of observing a spike count $k$ from neuron $n$ in state $i$ during an interval $\Delta t$ was given by the Poisson distribution with mean $\lambda_{ni}\Delta t$:

$$e_{ni}(k) = \frac{(\lambda_{ni}\Delta t)^k \, e^{-\lambda_{ni}\Delta t}}{k!}. \tag{6}$$

The neurons were assumed conditionally independent given the states, hence the emission probability in state $i$ and time bin $[t, t + \Delta t]$ ("bin $t$" for short) was the factorized distribution

$$e_i(\boldsymbol{k}(t)) = \prod_{n=1}^{N} \frac{(\lambda_{ni}\Delta t)^{k_n(t)} \, e^{-\lambda_{ni}\Delta t}}{k_n(t)!}, \tag{7}$$

where $k_n(t)$ is the spike count of neuron $n$ in bin $t$, $\boldsymbol{k}(t) = \{k_1(t), ..., k_N(t)\}$, and $t = 1, 2, ..., T$, where $T$ is the last bin in each trial. Note that the assumption of conditional independence given the states does not mean that the spike trains are treated as independent. Correlations between neurons are taken into account in the state transitions (see e.g. [7] for a good discussion of this point).

**Multinoulli-HMM (MHMM).** The MHMM is a categorical HMM with $N + 1$ observation symbols $\{1, 2, ..., N + 1\}$, where $N$ is the number of neurons. Symbol $n \leq N$ means that the $n$th neuron fires an action potential in the current bin, while no other neuron fires. Symbol $N + 1$ means that no neuron fires a spike in the current bin. No other observations are allowed, which requires $\Delta t$ to be very small (order of 1 ms, more on this later). When more than one neuron was found to emit a spike in the same time bin, we randomly selected one among them to be the firing neuron.

The observations were therefore a set of $N + 1$ discrete values that can be represented via one-hot encoding, i.e., observation $n$ can be encoded by the $N + 1$ dimensional vector $\{0, ..., 0, 1, 0, ..., 0\}$, where the only 1 occupies position $n$ (corresponding to neuron $n$ firing). The vector $\{0, 0, ..., 0, 1\}$ codifies the observation of zero spikes in the current bin. The observations in a given state $i$ obey therefore a finite discrete (or categorical) distribution $e_i(n)$, with $n = 1, 2, ..., N + 1$, for which we can use a 'multinoulli' representation [61, p. 35]

$$e_i(\boldsymbol{\delta}(t)) = \prod_{n=1}^{N+1} e_i(n)^{\delta_n(t)}, \tag{8}$$

where $\boldsymbol{\delta}(t) = \{\delta_1(t), ..., \delta_{N+1}(t)\}$ is the vector encoding the symbol observed in bin $t$ by one-hot encoding (i.e., $\delta_n = 1$ if symbol $n$ is observed, and $\sum_{n=1}^{N+1} \delta_n = 1$). The use of the name multinoulli-HMM (rather than categorical-HMM) reminds us of the way the observations are built: one neuron firing corresponds to a specific 'wire' being 'hot'. It is convenient to

parametrize $e_i(n)$ as follows:

$$e_i(n) = \begin{cases} p(\lambda_{ni}) \doteq 1 - e^{-\lambda_{ni}dt} & \text{if } n < N+1 \\ 1 - \sum_{j=1}^{N} p(\lambda_{ji}) & \text{if } n = N+1. \end{cases} \tag{9}$$

For very small binsdt and $n < N+1$, the parameter $\lambda_{ni}$ can be interpreted as the firing rate of neuron $n$ in state $i$. This is because $p(\lambda_{ni}) \approx \lambda_{ni}dt$ to leading order in $dt$, just as in a Poisson spike train with firing rate $\lambda_{ni}$.

A very short bin is also required for $e_i(n)$ to approximate the probability that symbol $n$ is observed from a set of $N$ Poisson spike trains. Under the Poisson assumption, the probability that neuron $n$ doesn't fire in a short time bin $[t, t+dt]$ is $1 - \lambda_{ni}dt$, hence the probability of observing symbol $n \leq N$ is approximately given by

$$\lambda_{ni}dt \prod_{\substack{j \neq n}}^{N+1} (1 - \lambda_{ji}dt) \approx \lambda_{ni}dt \, (1 - \mathcal{O}(dt)) \approx p(\lambda_{ni}), \tag{10}$$

where the last approximation holds to leading order in $dt$. Note that also in this case, in writing Eq 10, we have assumed that the neurons are conditionally independent given the state (whereas they are not in the multinoulli model). Note that it is still possible that two or more neurons spike in the same bin under the Poisson hypothesis – it is just less likely when the time bin is small. The requirement of having a very short time bin is therefore essential for the MHMM to be a good model of Poisson spike trains. We used time bins of 5 ms [15]. For larger time bins the MHMM, as an alternative way to model a Poisson HMM, will break down, especially for large numbers of neurons and/or large firing rates.

## Training the HMMs

Training an HMM is the procedure for estimating the parameters of the model given the data. We do so by using the Baum-Welch algorithm [1,3,7,62], which is a special case of the expectation-minimization algorithm [31]. The Baum-Welch algorithm updates the model parameters using re-estimation formulae that, at each step during training, increase (or at least do not decrease) the likelihood of the model given the data. The parameters of the model are the initial state probability vector $\pi$, the transition probability matrix $\Gamma$ and the emission probability matrix $E$, summarized by the symbol $\Theta = \{\pi, \Gamma, E\}$. The initial parameter values were selected as detailed in Sec. "Initial guess for the model parameters" below, and then re-estimated iteratively according to the Baum-Welch algorithm. We report here the main formulae of the algorithm while deferring full details to S2 Appendix.

The Baum-Welch algorithm varies slightly for different HMMs, however the following quantities will be needed in every case:

$$q_i(t) = P(S_t = i | O_{1:T}, \Theta), \tag{11}$$

i.e., the probability of state $i$ at time $t$, given the entire sequence of observations $O_{1:T}$ and the model parameters $\Theta$, and

$$\xi_{ij}(t) = P(S_t = i, S_{t+1} = j | O_{1:T}, \Theta), \tag{12}$$

the probability of a transition from state $i$ at time $t$ to state $j$ at time $t + 1$, given the observations and the model parameters. Note that

$$q_i(t) = \sum_{j=1}^{m} \xi_{ij}(t), \tag{13}$$

so that $q_i(t)$ is also the conditional probability of making a transition out of state $i$ at time $t$ (including a transition to $i$ itself).

We first present the algorithms assuming training on a single trial, deferring the case of multiple trials to a later subsection.

**PHMM.** For Poisson HMMs, the re-estimation formulae read (S2 Appendix)

$$\hat{\pi}_i = q_i(1), \qquad i = 1, ..., m \tag{14}$$

$$\hat{\gamma}_{ij} = \frac{\sum_{t=1}^{T-1} \xi_{ij}(t)}{\sum_{t=1}^{T-1} q_i(t)}, \qquad i, j = 1, ..., m \tag{15}$$

$$\hat{\lambda}_{ni}\Delta t = \frac{\sum_{t=1}^{T} q_i(t)k_n(t)}{\sum_{t=1}^{T} q_i(t)}, \qquad n = 1, ..., N; \quad i = 1, ..., m. \tag{16}$$

and the re-estimated emission probabilities $\hat{e}_i(\boldsymbol{k})$ are given by Eq 7 with $\lambda_{ni} = \hat{\lambda}_{ni}$.

The meaning of these formulae is that, at each iteration, the values on the right hand side are used to re-estimate $\pi_i, \gamma_{ij}$ and $\lambda_{ni}$ on the left hand side. This requires a current estimate of $q_i(t)$ and $\xi_{ij}(t)$, which is accomplished via the computation of auxiliary variables $\alpha_i$ and $\beta_i$ known as the forward and backward probabilities, respectively. The latter obey well known recursive relationships that make the re-estimation process tractable (see S2 Appendix, "The forward-backward algorithm"). In summary, the training algorithm alternates a step in which $q_i(t)$ and $\xi_{ij}(t)$ are computed based on the current estimate of the parameters $\pi_i, \gamma_{ij}$ and $\lambda_{ni}$ (called the 'expectation step', or E-step for short), and a step in which the parameters $\pi_i, \gamma_{ij}$ and $\lambda_{ni}$ are updated based on the current estimate of $q_i(t)$ and $\xi_{ij}(t)$ (called the 'maximization step', or M-step).

The forward probabilities allow also to compute $P(O_{1:T}|\Theta) = \sum_{i=1}^{m} \alpha_i(T)$, i.e., the likelihood of the model given the entire sequence of observations $O_{1:T}$. The Baum-Welch re-estimation algorithm guarantees that $P(O_{1:T}|\Theta)$ is not smaller than at the previous step, and therefore, if it converges, it will converge to a (local) maximum (or saddle) of the likelihood [30,63,64].

The PHMM was trained using custom code adapted from [3] (see Algorithm 1 for the pseudocode). To help prevent numerical overflow due to bins with zero spikes, we removed the neurons with maximum firing rate lower than 1 Hz across all trials. Moreover, during training, firing rates approaching zero were set to a very small positive value (0.001 spikes/s). The thresholds for convergence were set to $10^{-6}$ in all algorithms (parameters `tol`$_i$ in the pseudocode). The maximal number of iterations (`maxiter`) was set to 1,000 in all algorithms.

**MHMM.** The Baum-Welch re-estimation formulae for the MHMM are the same as for the Poisson HMM (with the observation at time $t$ being $\boldsymbol{\delta}(t)$ rather than $\boldsymbol{k}(t)$), except for the emission probabilities which, for symbol $n$ in state $i$, read (S2 Appendix, "M-step"):

$$\hat{e}_i(n) = \frac{\sum_{t=1}^{T} q_i(t)\delta_n(t)}{\sum_{t=1}^{T} q_i(t)} = \frac{\text{expected number of observations of symbol } n \text{ while in state } i}{\text{expected number of occurrences of state } i}, \tag{17}$$

**Algorithm 1. Poisson HMM** ($K_{N \times T}, \pi_{m \times 1}, \Gamma_{m \times m}, \Lambda_{N \times m}$, **dt, maxiter, tol**)

$K_{N \times T} = \{k_n(t)\}$ is the spike count matrix, $N$ is the number of neurons, $T$ is the number of time bins, $\pi_{m \times 1}$ is the vector of initial guesses for state probabilities, $m$ is the number of hidden states, $\Gamma_{m \times m}$ is the initial guess for the transition probability matrix, $\Lambda_{N \times m}$ is the initial guess for the firing rates matrix, $dt$ is the bin width ($\Delta t$ in Eq 7), $maxiter$ is the maximum number of iterations, $tol = \{tol_1, tol_2\}$ is a vector of tolerance levels for convergence.

```
 1: LL ← 1                                            ▷ log-likelihood ln P(O|Θ)
 2: for k = 1 to maxiter do
 3:     from K and Λ, compute the emission prob. matrix E_{N×T} with
        entries e_{it} = P(k(t)|Λ^i)                                   ▷ Eq 7
 4:     E step: compute forward and backward probabilities α, β
        from {π,Γ,E}                                              ▷ Eqs S2:19-22
 5:            compute LL_next, q_i, ξ_ij from α, β and {π,Γ,E}     ▷
        Eqs S2:23-25
 6:     M step: compute π_next, Γ_next, Λ_next              ▷ Eqs 14-16
 7:     if  ||Γ_next − Γ|| + ||Λ_next − Λ|| < tol_1  and  ||LL_next − LL|| < tol_2  then
 8:        Return π, Γ, Λ, LL                    ▷ training complete
 9:     else
10:         π ← π_next, Γ ← Γ_next, Λ ← Λ_next, LL ← LL_next  ▷ go to next iter.
        (line 2)
11:     end if
12: end for                                       ▷ end loop over iterations
```

with $\delta_n(t) = \{0, 1\}$ and $\sum_{n=1}^{N+1} \delta_n = 1$. Using the parametrization $e_i(n) = 1 - e^{-\lambda_{ni} dt}$, we can obtain the firing rates of the neurons in state $i$ from

$$\hat{\lambda}_{ni} = -\frac{1}{dt} \ln(1 - \hat{e}_i(n)), \tag{18}$$

but note that the parameters $\hat{\lambda}_{ni}$ play no role in the algorithm.

The MHMM was trained using Matlab's function HMMtrain, which trains a categorical HMM. The algorithm stops when non-appreciable differences ($<10^{-6}$) in the re-estimation of the transition probabilities, emission probabilities and the log-likelihood function are observed at the next step. The pseudocode of the training algorithm is given in Algorithm 2.

**sPHMM.** The sPHMM is a Poisson HMM that resets the self-transition probabilities above a threshold $\theta$ whenever it converges to a value smaller than $\theta$. The algorithm monitors the $\gamma_{ii}$ values and if one of them converges to a value $< \theta$ during training (see below), (i) it resets the transition matrix to its value at the most recent iteration time $t^*$ that met the requirement $\gamma_{ii} \geq \theta$ for all $i$; (ii) it restores the firing rate matrix to its value at $t^*$; and (iii) it shuffles the firing rate vectors across states (see below). Convergence to $\gamma_{ii} < \theta$ occurs when $|\gamma'_{ii} - \gamma_{ii}| < \epsilon$, where $\gamma'_{ii}$ is the value of $\gamma_{ii}$ at the next iteration and $\epsilon = 5 \cdot 10^{-5}$ was our tolerance level. Waiting for convergence ensures that the algorithm is not interrupted when the condition $\gamma_{ii} < \theta$ occurs only transiently; see Fig 12 for an example and Algorithm 3 for the pseudocode. Reshuffling the firing rates across states (pt. iii above) preserves the firing rates found by the algorithm until that point (assumed close to the true ones, thereby reducing the training time). In this paper we used $\theta = 0.8$ except for S1 Fig. The re-estimation formulae of the Baum-Welch algorithm are the same as for the PHMM. The pseudocode for the sPHMM is given in Algorithm 3.

**Algorithm 2. Multinoulli HMM** ($S_{T \times 1}, \pi_{m \times 1}, \Gamma_{m \times m}, E_{m \times (N+1)}$, **dt, maxiter, tol**)

Vector $S_{T \times 1}$ contains the sequence of symbols, $T$ is the number of time bins, $\pi_{m \times 1}$ is the vector of initial guesses for state probabilities, $m$ is the number of hidden states, $\Gamma_{m \times m}$ is the initial guess for the transition probability matrix, $E_{m \times (N+1)}$ is the initial guess for the emission probability matrix, $N$ is the number of neurons, dt is the bin width (see Eq 18), maxiter is the maximum number of iterations, tol = {tol₁, tol₂, tol₃} is a vector of tolerance levels for convergence.

*Note:* It requires a preprocessing stage that turns vectors of spike counts in bins into symbols.

```
 1: LL ← 1                                          ▷ log-likelihood P(O|Θ)
 2: for k = 1 to maxiter do
 3:    E step: compute forward and backward probabilities α, β
       from {π, Γ, E}                                ▷ Eqs S2:19-22
 4:          compute LL_next, q_i, ξ_ij from α, β and {π, Γ, E}   ▷
       Eqs S2:23-25
 5:    M step: compute π_next, Γ_next, E_next      ▷ Eqs 14, 15 and 17
 6:    if ||Γ_next − Γ|| < tol₁ and ||E_next − E|| < tol₂ and ||LL_next − LL|| < tol₃ then
 7:        compute estimated firing rates matrix Λ_est from E_next   ▷
       Eq 18
 8:        return π, Γ, E, LL, Λ                   ▷ training complete
 9:    else
10:        π ← π_next, Γ ← Γ_next, E ← E_next, LL ← LL_next   ▷ go to next iter.
       (line 2)
11:    end if
12: end for                                          ▷ end loop over iterations
```

**DPHMM.** In the DPHMM, we add a Dirichlet prior over the transition probabilities to nudge the self-transition probabilities towards high values. Since the rows of the transition matrix $\Gamma$ are probability distributions, i.e., $\gamma_{ij} \geq 0$ with $\sum_{j=1}^{m} \gamma_{ij} = 1$, the prior is added independently to each row $\Gamma_i$:

$$P(\Gamma_i | \boldsymbol{a}_i) = \frac{1}{B(\boldsymbol{a}_i)} \prod_{j=1}^{m} \gamma_{ij}^{a_{ij}-1}, \tag{19}$$

where the $a_{ij}$ are the parameters of the Dirichlet distribution, $\boldsymbol{a}_i = \{a_{i1}, ..., a_{im}\}$, and $B(\boldsymbol{a}_i)$ is a normalising factor (Beta function). We set $a_{ij} = 1.1$ for $i \neq j$ and $a_{ii} = 1 + 0.9(m-1)$, where $m$ is the number of hidden states (see below for explanation).

The quantities $a_{ij}$ control the amount of concentration of the $\gamma_{ij}$ around desired values. To encourage high values of the self-transition probabilities $\gamma_{ii}$, we must have relatively large $a_{ii}$ and smaller off-diagonal $a_{ij}$ values. Intuitively, this is because for large $a_{ii}$, $\gamma_{ii}^{a_{ii}-1}$ will be excessively small unless $\gamma_{ii} \approx 1$, and therefore large diagonal values are favored during the maximization of $P(\Gamma_i | \boldsymbol{a}_i)$. To choose appropriate values for $a_{ij}$, we looked at the mode of $P(\Gamma_i | \boldsymbol{a}_i)$; assuming $a_{ij} > 1$ for all $j$, the mode is a vector $\boldsymbol{\mu}_i$ with components

$$\mu_{ij} = \frac{a_{ij} - 1}{\sum_{k=1}^{m} a_{ik} - m}, \quad j = 1, ..., m. \tag{20}$$

The larger $a_{ii}$, the more concentrated the prior is on its mode, and the more this mode is characterized by a large $\gamma_{ii}$ and small $\gamma_{ij}$ for $j \neq i$. We therefore impose that the mode is equal to the desired $\gamma_{ii}$ value, below indicated with the symbol $\bar{\gamma}_{ii}$. Next, we choose the $a_{ij}$ for all $i \neq j$

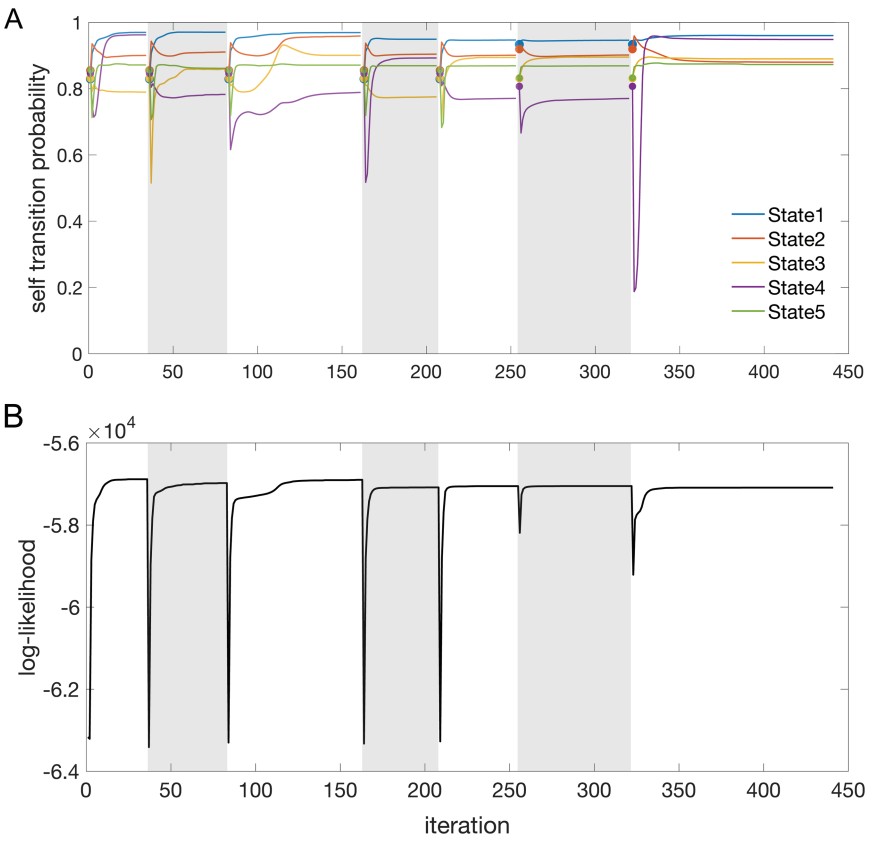

**Fig 12. Illustration of the training algorithm for the sPHMM. A:** Example of the temporal evolution of the self-transition probabilities $\gamma_{ii}$ (colored lines) for an HMM with five states (each state in a different color). The reset procedure was applied at the beginning of each block of iterations (white and grey bands). The values of the self-transition probabilities after each reset are marked by filled circles (see the text for details). **B:** Corresponding evolution of the model's log-likelihood during training.

to be equal to the same constant $\bar{a} > 1$, and solve Eq 20 for $a_{ii}$:

$$a_{ii} = 1 + \frac{\bar{\gamma}_{ii}(m-1)(\bar{a}-1)}{1-\bar{\gamma}_{ii}}. \tag{21}$$

This gives $a_{ii}$ as a function of $m$ given the parameters $\bar{\gamma}_{ii}$ and $\bar{a}$. We chose $\bar{\gamma}_{ii} = 0.9$ and $\bar{a} = 1.1$, giving $a_{ii} = 1 + 0.9(m-1)$. For example, for $m = 5$, this choice gives $a_{ii} = 4.6$, diagonal mode $\mu_{ii} = 0.9$, and non-diagonal modes $\mu_{ij} = 0.025$.

The re-estimation formulae of the Baum-Welch algorithm need only be slightly modified in this case. The only modification is in the update formula for $\hat{\gamma}_{ij}$, Eq 15, which is replaced by (see S2 Appendix)

$$\hat{\gamma}_{ij} = \frac{\sum_{t=1}^{T-1} \xi_{ij}(t) + a_{ij} - 1}{\sum_{t=1}^{T-1} q_i(t) + \sum_{j=1}^{m}(a_{ij} - 1)}. \tag{22}$$

Note that for an uninformative prior (all $a_{ij} = 1$) we recover the previous formula, Eq 15; otherwise, the re-estimation depends on the $a_{ij}$, which are fixed numbers chosen prior to

**Algorithm 3. Sticky Poisson HMM** ($\mathrm{K}_{\mathrm{N}\times\mathrm{T}}$, $\pi_{\mathrm{m}\times 1}$, $\Gamma_{\mathrm{m}\times\mathrm{m}}$, $\Lambda_{\mathrm{N}\times\mathrm{m}}$, $\theta$, `dt`, `maxiter`, `tol`)

$\mathrm{K}_{\mathrm{N}\times\mathrm{T}} = \{k_n(t)\}$ is the spike count matrix, $\mathrm{N}$ is the number of neurons, $\mathrm{T}$ is the number of time bins, $\pi_{\mathrm{m}\times 1}$ is the vector of initial guesses for state probabilities, $\mathrm{m}$ is the number of hidden states, $\Gamma_{\mathrm{m}\times\mathrm{m}}$ is the initial guess for the transition probability matrix, $\Lambda_{\mathrm{N}\times\mathrm{m}}$ is the initial guess for the firing rates matrix, $\theta$ is the threshold for the self-transition probabilities, `dt` is the bin width ($\Delta t$ in Eq 7), `maxiter` is the maximum number of iterations, `tol` = {$\mathrm{tol}_0$, $\mathrm{tol}_1$, $\mathrm{tol}_2$} is a vector of tolerance levels for convergence ($\mathrm{tol}_0$ is the tolerance for convergence prior to a reset).

1: $LL \leftarrow 1$ ▷ `log-likelihood` $P(O|\Theta)$

2: $\Gamma^* = \Gamma$, $\Lambda^* = \Lambda$ ▷ `store` $\Gamma$, $\Lambda$ `for reset`

3: **for** $k = 1$ to `maxiter` **do**

4: `from K and` $\Lambda$`, compute N × T emission prob. matrix E with entries` $e_{it} = P(\boldsymbol{k}(t)|\Lambda^i)$ ▷ Eq 7

5: **E step:** `compute forward and backward probabilities` $\alpha$, $\beta$ `from` $\{\pi, \Gamma, E\}$ ▷ Eqs S2:19–22

6: `compute` $LL_{next}$`,` $q_i$`,` $\xi_{ij}$ `from` $\alpha$`,` $\beta$ `and` $\{\pi, \Gamma, E\}$ ▷ Eqs S2:23–25

7: **M step:** `compute` $\pi_{next}$`,` $\Gamma_{next}$`,` $\Lambda_{next}$ ▷ Eqs 14–16

8: **if** $\Gamma_{next,ii} \geq \theta$ $\forall i = 1,...,m$ **then** ▷ `if all` $\gamma_{ii} \geq \theta$`, store current` $\Gamma$`,` $\Lambda$ `for next reset`

9: $\Gamma^* \leftarrow \Gamma_{next}$`,` $\Lambda^* \leftarrow \Lambda_{next}$

10: **else if** $\|\Gamma_{next} - \Gamma\| < \mathrm{tol}_0$ **then** ▷ `if one` $\gamma_{ii} < \theta$ `and` $\Gamma$ `has converged, reset` $\Gamma$`,` $\Lambda$

11: $\Gamma_{next} \leftarrow \Gamma^*$`,` $\Lambda_{next} \leftarrow \Lambda^*$

12: `shuffle` $\Lambda_{next}$ `across states`

13: **end if**

14: **if** $\|\Gamma_{next} - \Gamma\| + \|\Lambda_{next} - \Lambda\| < \mathrm{tol}_1$ `and` $\|LL_{next} - LL\| < \mathrm{tol}_2$ **then**

15: `Return` $\pi$`,` $\Gamma$`,` $\Lambda$`,` `LL` ▷ `training complete`

16: **else**

17: $\pi \leftarrow \pi_{next}$`,` $\Gamma \leftarrow \Gamma_{next}$`,` $\Lambda \leftarrow \Lambda_{next}$`,` $LL \leftarrow LL_{next}$ ▷ `go to next iter.` `(line 3)`

18: **end if**

19: **end for** ▷ `end loop over iterations`

running the EM algorithm. During training, the posterior

$$P(\Theta|O_{1:T}) \propto P(O_{1:T}|\Theta)P(\Theta), \tag{23}$$

rather than the likelihood, is monitored for convergence. The pseudocode is given in Algorithm 4.

**Training with multiple trials.** The algorithms presented so far apply when the datasets comprise a single trial. Our datasets are sessions comprising many trials. Assuming independent trials, in the re-estimation formulae for $\Gamma$ and $\Lambda$ (or $E$) one has to sum over all observations from each trial in both the enumerator and denominator (see e.g. Eq 16 of S2 Appendix). We give the pseudo-algorithm for the DPHMM in Algorithm 5. The coding strategy is analogous for the other HMMs, with the following changes:

– for the PHMM, remove line 20, replace line 18 with $(\Gamma_{next})_{ij} \leftarrow g_{ij}/g_i'$, and replace $LP$ with $LL$ in lines 3 and 21-24;

– for the MHMM, further remove line 8, replace the argument $\mathrm{K}_{\mathrm{N}\times\mathrm{T}\times\mathcal{T}}$ with $\mathrm{S}_{\mathrm{T}\times\mathcal{T}}$, replace $\Lambda$ with $E$ in lines 21-24, and replace line 19 with $(E_{next})_{ni} = l_{ni}/l_i'$. The latter gives $\hat{e}_i(n)$ of

**Algorithm 4. Poisson HMM with Dirichlet prior** ($\mathtt{K}_{\mathtt{N \times T}}, \pi_{\mathtt{m \times 1}}, \Gamma_{\mathtt{m \times m}}, \Lambda_{\mathtt{N \times m}}, \mathtt{A}_{\mathtt{m \times m}}, \mathbf{dt, maxiter, tol}$)

$\mathtt{K}_{\mathtt{N \times T}} = \{k_n(t)\}$ is the spike count matrix, $\mathtt{N}$ is the number of neurons, $\mathtt{T}$ is the number of time bins, $\pi_{\mathtt{m \times 1}}$ is the vector of initial guesses for state probabilities, $\mathtt{m}$ is the number of hidden states, $\Gamma_{\mathtt{m \times m}}$ is the initial guess for the transition probability matrix, $\Lambda_{\mathtt{N \times m}}$ is the initial guess for the firing rates matrix, $\mathtt{A}_{\mathtt{m \times m}} = \{a_{ij}\}$ is the matrix of parameters of the Dirichlet prior (Eq 19), $\mathtt{dt}$ is the bin width ($\Delta t$ in Eq 7), $\mathtt{maxiter}$ is the maximum number of iterations, $\mathtt{tol} = \{\mathtt{tol_1, tol_2}\}$ is a vector of tolerance levels for convergence.

```
1:  LP ← 1                                 ▷ total log-posterior ln P(O|Θ) + ln P(Θ)
2:  for k = 1 to maxiter do
3:      from K and Λ, compute the emission prob. matrix E_{N×T} with
        entries e_it = P(k(t)|Λ^i)                                   ▷ Eq 7
4:      E step: compute forward and backward probabilities α, β
        from {π, Γ, E}                                              ▷ Eqs S2:19-22
5:              compute LL_next, q_i, ξ_ij from α, β and {π, Γ, E}         ▷
        Eqs S2:23-25
6:      M step: compute Θ_next = {π_next, Γ_next, Λ_next}       ▷ Eqs 14, 22, 16
7:      LP_next ← LL_next + ln P(Θ)         ▷ Eq S2:29 with current value of Γ
8:      if ||Γ_next − Γ|| + ||Λ_next − Λ|| < tol_1  and  ||LP_next − LP|| < tol_2  then
9:          return π, Γ, Λ, LP                        ▷ training complete
10:     else
11:         π ← π_next, Γ ← Γ_next, Λ ← Λ_next, LP ← LP_next        ▷ iteration
        complete; go to line 3
12:     end if
13: end for                                            ▷ end loop over iterations
```

Eq 17, where $k_n(t)$ in line 15 now equals 1 if symbol $n$ has occurred in bin $t$, and zero otherwise;

– for the sPHMM, make the same changes as for the PHMM, and after line 19 continue from line 8 of Algorithm 3.

In the presence of multiple trials, if one can make the assumption that the data are collected under the same conditions, the initial state probability vector would be estimated as the average across trials. In the notation of Algorithm 5, this means $\pi = \mathcal{T}^{-1} \sum_r \pi^r$. When the trials come from different conditions (for example: in response to different stimuli), a single HMM cannot be a suitable representation of the data, but rather multiple HMMs (one for each condition) would be more appropriate. However, using a single HMM for segmentation purposes allows to compare hidden states across conditions, which is often of interest. To account for this situation, in Algorithm 5 we infer a separate initial state distribution for each trial. We note, however, that we assign no importance to the initial state distributions except for their use in the re-estimation formulae – i.e., we treat them as auxiliary variables of the training algorithm, and consider the HMMs fully characterized by the transition and emission probabilities. An alternative approach is to fix the initial state distribution without re-estimating. This is accomplished in Algorithm 5 by removing line 12.

**Initial guess for the model parameters.** In the MHMM, the initial transition probability matrix $\Gamma$ was a random symmetric matrix with diagonal elements close to 1 and normalized rows. Specifically,

$$\Gamma_{init} = \theta \mathbf{I}_m + (1 - \theta) U, \tag{24}$$

where $\theta = 0.9$, $\mathbf{I}_m$ is the $m \times m$ identity matrix, and $U$ is a matrix with elements $u_{ij}/\sum_{j=1}^{m} u_{ij}$, where $u_{ij}$ is a uniform random variable between 0 and 1 and $u_{ji} = u_{ij}$. The emission probability matrix $E$ was randomly chosen with the values in the first column being very close to 1, and normalized rows (i.e., $\sum_{n=1}^{N+1} e_i(n) = 1$). The initial state distribution was set to $\pi_i = \delta_{i1}$.

**Algorithm 5. DPHMM with multiple trials** ($K_{N \times T \times \mathcal{T}}$, $\pi_{m \times 1}$, $\Gamma_{m \times m}$, $\Lambda_{N \times m}$, $A_{m \times m}$, `dt`, `maxiter`, `tol`)

$K_{N \times T} = \{k_n(t)\}$ is the spike count matrix (one for each trial), N is the number of neurons, T is the number of time bins in each trial, $\mathcal{T}$ is the number of trials, $\pi_{m \times 1}$ is the vector of initial guesses for state probabilities, m is the number of hidden states, $\Gamma_{m \times m}$ is the initial guess for the transition probability matrix, $\Lambda_{N \times m}$ is the initial guess for the firing rates matrix, $A_{m \times m} = \{a_{ij}\}$ is the matrix of parameters of the Dirichlet prior (Eq 19), dt is the bin width ($\Delta t$ in Eq 7), `maxiter` is the maximum number of iterations, `tol` = {$\text{tol}_1$, $\text{tol}_2$} is a vector of tolerance levels for convergence.

```
 1: maxtrials = 𝒯              ▷ extract total number of trials from K
 2: πʳ ← π for every trial r; Π ← [π¹,...,π𝒯]   ▷ Π_{m×𝒯} = matrix with
    elements πᵢʳ
 3: LP ← 1                      ▷ total log-posterior ln P(O|Θ) + ln P(Θ)
 4: for k = 1 to maxiter do
 5:     LL_next = 0
 6:     G_{m×m}, G'_{m×1}, L_{N×m}, L'_{m×1} = 0   ▷ G = matrix with elements gᵢⱼ, etc.
    Needed in lines 13-19
 7:     for r = 1 to maxtrials do    ▷ r indexes the current trial
 8:         from K and Λ, compute current trial's matrix Eʳ_{N×T} with
    entries eᵢₜ = P(k(t)|Λⁱ)                          ▷ Eq 7
            Eʳ step:
 9:         compute αʳ, βʳ from {πʳ,Γʳ,Eʳ}           ▷ Eqs S2:19-22
10:         compute LLʳ, qʳ, ξʳ from αʳ, βʳ and {πʳ,Γʳ,Eʳ}   ▷
    Eqs S2:23-25
11:         LL_next ← LL_next + LLʳ        ▷ update the total likelihood
            Mʳ step:
12:         πʳ_next = qʳ(1)                     ▷ Eq 14 (one for each trial)
13:         gᵢⱼ ← gᵢⱼ + Σᵀ⁻¹_{t=1} ξʳᵢⱼ(t)    ▷ to build numerator of Eq 22
14:         g'ᵢ ← g'ᵢ + Σᵀ⁻¹_{t=1} qʳᵢ(t)    ▷ to build denumerator of Eq 22
15:         lₙᵢ ← lₙᵢ + Σᵀ_{t=1} qʳᵢ(t)kₙ(t)   ▷ to build numerator of Eq 16
16:         l'ᵢ ← l'ᵢ + Σᵀ_{t=1} qʳᵢ(t)        ▷ to build denumerator of Eq 16
17:     end for                              ▷ end loop over trials
        M step:
18:     (Γ_next)ᵢⱼ ← (gᵢⱼ + aᵢⱼ - 1)/(g'ᵢ + Σᵐ_{j=1}(aᵢⱼ - 1))   ▷ make sure to normalize
    over j
19:     (Λ_next)ₙᵢ ← lₙᵢ/l'ᵢ
20:     LP_next ← LL_next + ln P(Θ)       ▷ Eq S2:29 with current value of Γ
21:     if ||Γ_next - Γ|| + ||Λ_next - Λ|| < tol₁ and ||LP_next - LP|| < tol₂ then
22:         return Π, Γ, Λ, LP                ▷ training complete
23:     else
24:         Π ← Π_next, Γ ← Γ_next, Λ ← Λ_next, LP ← LP_next         ▷ iteration
    complete; go to line 4
25:     end if
26: end for                              ▷ end loop over iterations
```

In the Poisson HMMs, $\Gamma_{init}$ was given by Eq 24 with $\theta = 0.8$, while the initial neuron firing rates $\lambda_{ni}$ were sampled from a uniform distribution between the minimal and maximal firing rates estimated from the data used for training. The initial state distribution was set to $\pi_i = 1/m$, where $m$ was the number of hidden states.

For each $m$, we trained the HMMs with 100 different sets of initial guesses for the model parameters (each chosen as described above), except for the analysis of Fig 7, where we used

100 and 1,000 sets of initial guess, and Fig 11, where we used between 10 and 1000 sets of initial guesses.

We have also explored the strategy of clustering the ensemble firing rate vectors to infer the initial firing rates. We used two main methods. In the first method, we used clusters of firing rate vectors estimated from the peri-stimulus time histogram (PSTH). For each neuron, we computed the PSTH of all the trials in the training set, using the same time bin used for training the HMM. For each bin, we got a vector of spike counts across the $N$ neurons. We then clustered the spike count vectors into $m$ clusters, and used the centroids of each cluster to build the $m$ columns of the initial firing rate matrix $\Lambda_{N \times m}$ (after converting spike counts into firing rates). Alternatively, instead of the PSTH, in the method above we used the PSTH smoothed with a Gaussian kernel.

In the second method, for a given $m$, we first trained an sPHMM with $\tilde{m} > m$ states; the firing rate vectors decoded in each bin were clustered into $m$ clusters. The centroids of the clusters were then used to obtain the initial firing rate matrix.

For both methods, the clustering algorithms used were $K$-means and DBSCAN [44]. $K$-means was run each time with 10 different set of initial guesses (random assignments of the initial centroids) to select the best clustering model. The latter was used to build the initial firing rate matrix for the HMM.

## Model selection

Model selection was based on the maximum likelihood principle, i.e., the best model was the model with the highest likelihood given the data. We used three methods: cross-validation, BIC and AIC, described below.

**Cross-validation.** For cross-validation, we split each dataset in 5 non-overlapping sets (folds). For each fold, we trained the HMM on 4/5 of the data and tested the model on the remaining fold (validation set). The number of states $m$ was set to a value between 2 to $M = 20$ (25 in Fig 7). For each $m$, we trained the HMM multiple times, using each time a different set of random guesses for the initial parameter values as described in Sec. "Initial guess for the model parameters". For each set of initial guesses, and given $N_v$ trials in the validation set, we computed the LL of the model on the validation set, $LL_v(m)$, according to

$$LL_v(m) = \sum_{k=1}^{N_v} LL_k(m), \tag{25}$$

where $LL_k$ is the log-likelihood of the model in trial $k$ (we assume that different trials are independent and therefore the probabilities multiply). The average $LL_v(m)$ across folds and initial guesses, $\langle LL_v(m) \rangle$, was then compared for different $m$. We used three methods to derive the optimal number of states, $m^*$: in the first method ($CV_{max}$), we chose the point for which $\langle LL_v(m) \rangle$ had a maximum; in the second method ($CV_{slope}$), $m^*$ was chosen as the point of largest change in slope of the $\langle LL_v(m) \rangle$ curve; in the third method ($CV_{1SD}$), we selected the minimal value of $m$ for which one SD above $\langle LL_v(m) \rangle$ is larger than one SD below $\langle LL_v(m_{max}) \rangle$. Given $m^*$, the model that had the largest $LL_v(m^*)$ across initial guesses for the parameter values was chosen as the best model. We found that the average cross-validated LL gave the same results as the LL in single folds, which for better clarity are shown in the figures.

For the DPHMM, we used an additional selection criterion, i.e., we chose $m^*$ that maximized $\langle LP_v(m) \rangle = \langle LL_v(m) + \sum_{i=1}^{m} \sum_{j=1}^{m} (a_{ij} - 1) \ln \hat{\gamma}_{ij} \rangle$, a quantity that differs from the

average log-posterior of $\Theta$ by a term that does not depend on the model parameters (see S2 Appendix).

**BIC and AIC.**   Procedures based on BIC and AIC used the entire dataset as the training set, but penalized the log-likelihoods according to the number of parameters to be tuned (so that more complex models are penalized compared to simpler ones). The model with the largest penalized likelihood on the training set conventionally corresponds to the minimum of BIC or AIC; for $m$ states, these were defined as

$$\mathrm{BIC}(m) = -2LL_t(m) + K \ln D \qquad (26)$$

$$\mathrm{AIC}(m) = -2LL_t(m) + 2K, \qquad (27)$$

where $K$ is the number of parameters tuned during training and $D$ is the total number of observations in each session (i.e., the total number of time bins across all trials). $LL_t(m)$ is the same quantity as in Eq 25, except evaluated on the training set (after training). For each $m$, the BIC and AIC scores were averaged across the initial guesses for the parameter values; the optimal number of states $m^*$ was chosen as the value for which the mean BIC (or AIC) had a minimum. The model with $m^*$ hidden states that had the lowest score across initial guesses was chosen as the best model.

The number of parameters $K$ was given by the number of independent parameters required to determine the transition ($\Gamma$) and emission probability matrices. $\Gamma$ is an $m \times m$ matrix with rows summing to 1, hence having $m(m-1)$ degrees of freedom. The emission probabilities are determined by the firing rates of $N$ neurons in $m$ different states, giving $mN$ parameters, and thus

$$K = m(m-1) + mN. \qquad (28)$$

This formula applies to both the PHMM and the MHMM (in the latter, the emission probability matrix has $m(N+1)$ elements, but the rows sum to 1, removing $m$ parameters). The initial state probabilities were treated as auxiliary variables during training (see Sec. "Training with multiple trials") and did not contribute to $K$.

The modified BIC and AIC scores plotted in Fig 4 (green lines) were obtained by replacing $LL_t(m)$ with the log-posterior $LP_t(m) = LL_t(m) + \sum_{i=1}^{m} \sum_{j=1}^{m} (a_{ij} - 1) \ln \hat{\gamma}_{ij}$ in Eqs 1-2 (see S2 Appendix). For BIC, this comes from the asymptotic expansion of the log-posterior probability of the parameters given the data. This is given by $LL_t(m) + \ln P(\hat{\theta}) + \frac{K}{2} \ln 2\pi - \frac{1}{2} \ln |A|$ (see e.g. [65, p. 216]), where $|A|$ is the determinant of $A$, the Hessian matrix of the log-posterior distribution, and the log-prior $\ln P(\hat{\theta}) = \sum_{i=1}^{m} \sum_{j=1}^{m} (a_{ij} - 1) \ln \hat{\gamma}_{ij}$ + irrelevant terms. Note that the log-prior term is of order $m^2$ and therefore of the same order as $K \ln 2\pi$; therefore one should retain this term together with the log-prior. Since $K \ln 2\pi$ and the log-prior have different signs, the modified BIC would lay between the green and the black curves in Fig 4, leaving the results unchanged. Adding the log-prior to the AIC is more difficult to justify; here, we take the heuristic approach of taking the AIC as an estimate of the in-sample error where the loss function is given by the negative log-posterior rather than the negative likelihood, resulting in $-2LP_t(m)$ as the training error; see e.g. ref. [24,Sec. 7.5], for a discussion of this topic.

## Decoding the HMMs

Decoding is the procedure of inferring the hidden states from a sequence of observations and according to an HMM with parameters $\Theta$ (in our case, the best model selected after training). Data were divided into bins of the same length used for training. In each bin, the state having

the maximum posterior probability given the observations and the model, was decoded as the hidden state; that is:

$$S_t = \arg\max_i P(S_t = i | O_{1:T}, \Theta) = \arg\max_i q_i(t), \tag{29}$$

where $q_i(t)$ is given by Eq 24 of S2 Appendix. Bins where $P(S_t = i | O_{1:T}, \Theta) < 0.8$ for all $i$ were not assigned to any state, i.e., the decoding was undecided in those bins. The initial state probability $\pi$ was set equal to the stationary distribution, given by the solution to $\pi^\top (\mathbf{I}_m - \Gamma + \mathbf{1}_m) = 1_m$, where $\pi^\top$ is the transpose of $\pi$, $\mathbf{I}_m$ is the $m \times m$ identity matrix, $\mathbf{1}_m$ is the $m \times m$ matrix of ones, and $1_m$ is a row vector of ones (see e.g. [3], p. 19]). An equivalent approach is to use $\pi_i = \delta_{i1}$ (always start from the first state), but begin decoding a few hundred ms prior to the period of interest [13].

An alternative decoding method assigns states to each bin so as to maximize the probability of the whole sequence (Viterbi decoding; see e.g. [1]). In this decoding scheme, each bin is assigned a state (no undecided bins), yet this does not prevent rapid state switching, as shown in Fig 1. Viterbi decoding in also implemented in the accompanying code.

## Model comparison

The comparison index $\rho$ between two HMMs was defined as

$$\rho(\Theta_1, \Theta_2) = \frac{D(\Theta_1)}{D(\Theta_2)}, \tag{30}$$

where $\Theta_i = (\Lambda_i, \Gamma_i)$ are the parameters of trained HMM models that need comparison. $D(\Theta)$ was defined as the sum of the squared differences between the observed and inferred spike counts in each bin:

$$D(\Theta) = \sum_{k=1}^{K} \sum_{t=1}^{T_k/dt} \sum_{n=1}^{N} (X_k(n,t) - f_{\Theta,k}(n,t)\, dt)^2, \tag{31}$$

where $X_k(n,t)$ is the spike count of neuron $n$ in time bin $t$ in trial $k$ (lasting $T_k$) while $f_{\Theta,k}(n,t)$ is the model-inferred firing rate of the same neuron in the same bin, obtained from the decoded state in that bin. Each bin was decoded using posterior decoding, assigning the bin to the state with the largest posterior probability in that bin. $D(\Theta)$ was always computed on a validation set not used for training.

$D(\Theta)$ quantifies the irreducible variability of the data under the assumption that it was generated by an HMM with parameters $\Theta$. It does so by combining information about both the firing rate vectors defining the hidden states and the state transitions. If two models have the same states, the same state transitions, and the same transition times on a given dataset, we expect $\rho \approx 1$. When a true model is available, we set $\Theta_2 = \Theta_{true}$ and expect $\rho \geq 1$. In that case, $D(\Theta_{true})$ is computed after decoding the data with the model used to generate it (to account for the variability due to state transitions occurring inside a bin).

To compare and best-match the states of two HMMs we used the so-called Hungarian algorithm [29] as implemented by Matlab's function `matchpairs`. The algorithm matches the states of two different models according to their similarity: state $i$ of model $A$ is matched to state $j$ of model $B$ if the Euclidean distance $d(i,j)$ between the corresponding firing rate vectors is lowest across all distances $d(a,b)$ with $a \in A, b \in B$. After running this procedure, the states were relabeled so that the matched states are given the same label.

## Supporting information

**S1 Fig. Threshold vs. bin width in the sPHMM. A:** sPHMM decoding of an EXP dataset with 9 neurons after training with three different sets of (dt, $\theta$) values which correspond to a mean state duration of 250 ms. The three sets of parameters were, from top to bottom: ($dt = 100\ ms, \theta = 0.6$); ($dt = 50\ ms, \theta = 0.8$); ($dt = 25\ ms, \theta = 0.9$). **B:** The hidden states corresponding to the results shown in panel A (same color code). Each panel shows the inferred firing rates for each state for the three different sets of parameters. No appreciable difference is visible across states. The average Euclidean distance among corresponding states is 0.9 spikes/s and is less than 2.6 spikes/s in all cases.
(PDF)

**S2 Fig. $\rho$ values vs. initial parameter guesses.** $\rho = D(\Theta)/D(\Theta^*)$ for the models in Fig 11 of the main manuscript, where $\Theta^*$ is the best model trained with 1,000 initial conditions (*in lieu* of the true model). **A:** Histogram of the $\rho$ values when $\Theta$ is the model trained with 100 initial conditions (blue) compared with the histogram of the same model after shuffling its states' firing rates and off-diagonal transition probabilities (100 shuffles for each trained model). The $\rho$ values were narrowly distributed around 1, with a probability of 0.0082 of getting the largest $\rho$ value (or a smaller value) under the shuffled model. **B:** Same as panel A with $\Theta$ being the model trained with 1,000 initial conditions. The probability of getting the trained $\rho$ values under the shuffled model is 0.0098.
(PDF)

**S1 Appendix. State persistence and self-transition probabilities**
(PDF)

**S2 Appendix. The expectation-maximization algorithm for HMMs**
(PDF)

## Acknowledgments

We are indebted to Dr. Alfredo Fontanini for sharing the experimental datasets analyzed in this work. We thank Drs. Christian Quaia, Memming Park, Braden Brinkman, Alfredo Fontanini, Paul Miller and Brent Doiron for useful discussions, and Dr. Christian Quaia for technical exchanges on HMMs. This work was partially supported by an U01 grant from the NIH/NINDS Brain Initiative (1UF1NS115779).

## Author contributions

**Conceptualization:** Giancarlo La Camera.

**Formal analysis:** Tianshu Li.

**Funding acquisition:** Giancarlo La Camera.

**Project administration:** Giancarlo La Camera.

**Resources:** Giancarlo La Camera.

**Software:** Tianshu Li.

**Supervision:** Giancarlo La Camera.

**Writing –– original draft:** Tianshu Li, Giancarlo La Camera.

**Writing –– review & editing:** Tianshu Li, Giancarlo La Camera.

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
