## [Decision Letter · Decision Letter 0]

8 Jan 2025

PONE-D-24-50170A sticky Poisson Hidden Markov Model for solving the problem of over-segmentation and rapid state switching in cortical datasetsPLOS ONE

Dear Dr. La Camera,

Thank you for submitting your manuscript to PLOS ONE. After careful consideration, we feel that it has merit but does not fully meet PLOS ONE’s publication criteria as it currently stands. Therefore, we invite you to submit a revised version of the manuscript that addresses the points raised during the review process.

There is good agreement between the reviewers, particularly as regards the need to clarify a few technical things (e.g., the threshold, number of states). In addition, considering the wide audience of PLoS One, I would urge the authors to heed the recommendation of one of the reviewers to improve the presentation so that all readers (including the targeted ones) can fruitfully engage with the material. 

We look forward to receiving your revised manuscript.

Kind regards,

Luc Berthouze

Academic Editor

PLOS ONE

Journal Requirements:

NIH/NINDS Brain Initiative (1UF1NS115779) to G.L.C.  

We are indebted to Dr. Alfredo Fontanini for sharing the experimental datasets analyzed in this work. We thank Drs. Christian Quaia, Memming Park, Braden Brinkman, Alfredo Fontanini, Paul Miller and Brent Doiron for useful discussions, and Dr. Christian Quaia for technical exchanges on HMMs. This work was partially supported by an U01 grant from the NIH/NINDS Brain Initiative (1UF1NS115779) to G.L.C.

NIH/NINDS Brain Initiative (1UF1NS115779) to G.L.C.

Reviewers' comments:

Reviewer's Responses to Questions

**Comments to the Author**

1. Is the manuscript technically sound, and do the data support the conclusions?

Reviewer #1: Yes

Reviewer #2: Yes

2. Has the statistical analysis been performed appropriately and rigorously? 

Reviewer #1: Yes

Reviewer #2: Yes

3. Have the authors made all data underlying the findings in their manuscript fully available?

Reviewer #1: Yes

Reviewer #2: Yes

4. Is the manuscript presented in an intelligible fashion and written in standard English?

Reviewer #1: Yes

Reviewer #2: Yes

5. Review Comments to the Author

Reviewer #1: PLoS SHMM Review Dec 2024

The paper is in most ways thorough and clear, with good data backing up the proposed methods for producing optimal HMMs of data. There are a few things I would like to see mentioned minimally, and ideally carried out as time permits:

1) While the authors mention a threshold other than 0.8 could be used, I think it important to connect the threshold to the time-bin being used for the model. For example, with 5ms bins a minimum of 0.8 for self-transitions means a maximum of a 1/5 probability per 5ms of a transition. If time bins were 10ms the minimum would be about 0.64 and with 1ms time bins, on the order of 0.96 (approx. 1/25 probability per time bin). This should be discussed and ideally a demonstration that the method with altered time bin and correspondingly altered threshold leads to the same results, whereas a very different time bin but unaltered threshold would be less ideal.

2) Throughout the authors use the measure of number of states extracted as a measure of the validity of the model. Whereas the model produces firing rates as a function of time on each trial. It would be nice to show that the “correct” no. of states better matches these rates vs time when known – or how well transition times are captured.

3) Similarly, when looking at the number of initial conditions, in such a situation the models being compared have the same number of states so LL is a good measure of the accuracy of the model. It would be nice to see a histogram of LLs achieved from different initial conditions, or to see the typical number of times the same optimal state is reached with different numbers of initial conditions. That is, when the authors say they see little improvement moving to 1000 i.c.s, is that because the best model found with 100 i.c.s is now reached 10 or more times, or because different models with similar LL are reached? Just focusing on how often the “correct no. of states in the model” is found leaves out plenty of info. That is, models that reach the same no. of states can find very different sets of states and probabilities, so how accurate are those sets of states, and how often are they the same across i.c.s?

4) P. 16, “a subset of states were … difficult to separate”: this can be quantified. Each time the state is visited there is a measured firing rate (number of spikes divided by duration) that is different from the fiducial/veridical firing rate for each cell. One could look at the distributions of these data points and see if the clusters overlap for different states and mention their separability (this has been used to determine optimal number of states in some prior works – they stop adding states when there is overlap in distributions).

Minor comments:

First line of intro: “random transitions” – “randomly timed” is better as the structure of the HMM shows that they are constrained to where they can go, sometimes in a reliable sequence. (The word “random” without a qualifier suggests equally likely to go to any state).

Fig 2 caption “on representative of all three” has a grammar error.

p.7 bottom. “But how to … strategy?” is not a sentence

p. 8 “suggest to use” is incorrect grammar

Reviewer #2: From what I understand, the paper basically implements a minimum threshold on self-transition elements of the transition probability matrix to solve the problem of overfitting when applying HMMs to neural data. With my background in applying HMMs to fluorescence microscopy data, I was able to mostly understand the paper however the presentation can improve a bit to make it more pedagogical. At the moment, I suspect more biology oriented audience would have hard time understanding everything in the paper. I have following addiitional concerns.

1. In Fig 1. and others, it is visually hard to understand the color coding of states. The caption is not clear about what the colors represent in the context of HMM. I am guessing they represent one of the possible states of the system. In the caption, they should be connected to the m-dimensional state space mentioned in the text.

2. Page 3, Intro section, second paragraph: I am confused about the number of hidden states being called the order of HMM. Usually, in the physics literature I have read, order refers to how many hidden states in the time sequence of hidden states the observations depend on. For instance, in the simplest single-order HMM, an observation w_n at time t_n would depend on the state s_n only, that is, a memory-less process. In a second-order HMM, w_n may depend on two states like s_{n-1} and s_n. Not a memory less process. Am I missing something?

3. The authors come up with the value of 0.8 for the minimum threshold. The physical reasoning for this choice should be mentioned the first time this value it appears and not until much later in the Appendix A.

4. I am also concerned about the mathematical innovation in this paper. Most of the techniques used in the paper, AIC, BIC, sticky priors, et cetera are well established in the literature even in biophysics community. This means the paper is mostly about the application of these techniques to neural dynamics. So, the paper would meet the criteria for publication as long as the application is novel/important enough, which hopefully other reviews will comment on.

5. It would be nice if the authors can comment on other nonparametric HMM methods such as infinite Hidden Markov Models (iHMM) with unknown number of states for the application in question. They do some of this in the discussion but additional comments would be helpful.

6. PLOS authors have the option to publish the peer review history of their article (what does this mean?). If published, this will include your full peer review and any attached files.

Reviewer #1: No

Reviewer #2: No

---

## [Editor Report · Decision Letter 1]

22 May 2025

A sticky Poisson Hidden Markov Model for solving the problem of over-segmentation and rapid state switching in cortical datasets

PONE-D-24-50170R1

Dear Dr. La Camera,

We’re pleased to inform you that your manuscript has been judged scientifically suitable for publication and will be formally accepted for publication once it meets all outstanding technical requirements.

Kind regards,

Luc Berthouze

Academic Editor

PLOS ONE

Additional Editor Comments (optional):

Thank you for thoroughly engaging with the review process.

---

## [Editor Report · Acceptance letter]

PONE-D-24-50170R1

PLOS ONE

Dear Dr. La Camera,

I'm pleased to inform you that your manuscript has been deemed suitable for publication in PLOS ONE. Congratulations! Your manuscript is now being handed over to our production team.

Kind regards,

on behalf of

Prof. Luc Berthouze

Academic Editor

PLOS ONE